# RTL-OPT: Rethinking the Generation of PPA-Optimized RTL Code and A New Benchmark

## Abstract

The rapid advancements of AI rely on the support of integrated circuits (ICs). Recently, large language models (LLMs) have been increasingly explored in the generation of IC designs, mostly in Register-Transfer Level (RTL) code format, such as Verilog or VHDL. However, most existing benchmarks focus primarily on the accuracy of RTL code generation, rather than the optimization of IC design quality in terms of power, performance, and area (PPA). This work critically examines RTL optimization benchmarks and highlights the challenges of assessing RTL code quality. Our findings show that optimization assessments are complex and existing works yield misleading results, as the perceived superiority of RTL code often depends on the downstream synthesis tool and setup. To address these issues, we introduce RTL-OPT, a benchmark comprising 36 digital IC designs handcrafted by our human designers. These designs incorporate diverse optimization patterns derived from proven industry-standard RTL practices. Such optimization opportunities are not utilized by automated downstream logic synthesis, making them meaningful RTL code improvements. In addition, RTL-OPT covers a wide range of RTL implementation types, including combinational logic, pipelined datapath, finite-state machines, and memory interfaces, making it sufficiently representative. For each design task, RTL-OPT provides a pair of RTL codes: a carefully designed suboptimal (i.e., to-be-optimized) RTL code and an optimized RTL code as the golden reference. LLMs are expected to take the suboptimal RTL code as input, then generate a more optimized RTL code that leads to better ultimate PPA quality. The golden references, as a comparison baseline, reflect optimizations at the human-expert level. RTL-OPT further provides an integrated evaluation framework to automatically verify functional correctness and quantify PPA improvements of the LLM-optimized RTL code. This framework enables a standardized assessment of generative AI's ability in hardware design optimization. RTL-OPT is available at https://anonymous.4open.science/r/RTL-OPT-20C5.

## 1 Introduction

The rapid advancements of AI rely on the support of integrated circuits (ICs), which are increasingly complex and difficult to optimize. In recent years, the adoption of Large Language Models (LLMs) in the agile design of ICs has emerged as a promising research direction (Fang et al., 2025). Especially, many recent works (Liu et al., 2024b; Ho et al., 2024; Liu et al., 2023a; Pei et al., 2024; Fu et al., 2023; Chang et al., 2023; Thakur et al., 2023; Cui et al., 2024; Liu et al., 2024a; 2025; Zhao et al., 2024) develop customized LLMs to directly generate IC designs in the format of Register-Transfer Level (RTL) code, such as Verilog or VHDL.

**Benchmarking RTL Code Generation.** The RTL design is the starting point of digital IC design implementation and requires significant human efforts and expertise. LLM-assisted RTL code generation techniques (Liu et al., 2024b; Ho et al., 2024; Liu et al., 2025; Zhao et al., 2024) aim to relieve engineers from the tedious RTL coding process. To enable a fair comparison among different LLMs' capabilities in RTL generation, high-quality benchmarks become necessary. Representative benchmarks on RTL code generation include VerilogEval (Liu et al., 2023b) and RTLLM (Lu et al., 2024), VerilogEval v2 (Pinckney et al., 2024), RTLLM 2.0 (Liu et al., 2024c), CVDP (Pinckney et al., 2025), and others (DeLorenzo et al., 2024; Allam & Shalan, 2024).

**Limitation: Lack of Optimization.** However, the aforementioned RTL generation benchmarks primarily focus on the *correctness* of RTL code generation, without explicitly evaluating the *optimization of IC design's ultimate qualities* in terms of power, performance, and area (PPA). Such PPA quality is a unique property of hardware RTL code, in comparison with software code. In hardware design, the RTL code will be synthesized into ultimate circuit implementations using synthesis tools. Similar to a compiler in software, synthesis tools will apply extensive logic optimizations when converting RTL code into implementations. Thus, PPA results depend on both the RTL code quality and the downstream synthesis process. As we will point out in this paper, this tight interplay makes benchmarking RTL optimization particularly challenging, sometimes even misleading.

**Benchmarking RTL Code Optimization.** Most recently, some LLM works (Yao et al., 2024; Wang et al., 2025; Xu et al., 2025) start to target generating more *optimized* RTL code, which is expected to yield better ultimate chip quality in PPA. These works are all evaluated on the only relevant benchmark (Yao et al., 2024), which provides sub-optimal RTL codes for LLMs to improve. However, our study indicates that this benchmark (Yao et al., 2024) falls short in several aspects: 1) **Unrealistic designs:** many sub-optimal RTL codes in this benchmark are overly contrived and fail to capture real inefficiencies in practice; 2) **Oversimplified synthesis setup:** reliance on weak synthesis tools such as Yosys (Wolf et al., 2013) leads to results that are sensitive to superficial RTL code changes and poorly aligned with industrial-grade flows; 3) **Insufficient evaluation:** its assessments focus only on area-related metrics, while neglecting power and timing. Such evaluation metric neglects the ubiquitous trade-offs in a typical IC design process.

In this work, we first inspect the existing works on RTL optimization and rethink a key question: **how to benchmark the optimization of RTL code appropriately?** We carefully inspect existing works and downstream synthesis flows. This study reveals that evaluating RTL optimization is non-trivial and may easily lead to misleading conclusions. Specifically, whether one RTL code is superior (i.e., more optimized) to the other strongly depends on the synthesis tool and setup. Many "optimized" RTL codes indicated by the prior work (Yao et al., 2024) turn out to be the same or even worse than their "sub-optimal" RTL counterparts when different, typically more advanced, synthesis options are adopted.

Based on our aforementioned observations, we propose a new benchmark, **RTL-OPT**, specifically designed to systematically evaluate LLMs' ability in RTL design optimization. RTL-OPT consists of 36 handcrafted RTL optimization tasks targeting PPA qualities. A key distinguishing feature is that it provides a collection of **diverse and realistic optimization patterns**, such as *bit-width optimization, precomputation and LUT conversion, operator strength reduction, control simplification, resource sharing, and state encoding optimization*, all derived from proven industry practices. These patterns capture transformations that truly matter for RTL optimization and remain effective even under advanced synthesis. It sets RTL-OPT apart from prior works that often lacked real optimization impact. As illustrated in Figure 1, each task in RTL-OPT provides a pair of RTL codes: a deliberately designed sub-optimal (to-be-optimized) version and an optimized version serving as the golden reference. Any benchmarked LLM takes the sub-optimal code as input and attempts to generate a more optimized RTL code while preserving design functionality. Specifically, RTL-OPT provides: 1) a set of 36 handcrafted tasks, ensuring comprehensive and representative coverage of real-world design challenges; 2) an integrated evaluation framework (Figure 1), which automatically verifies functional correctness and compares the ultimate PPA of LLM-optimized designs against the designer-optimized golden reference.

Constructing a high-quality benchmark for RTL optimization is inherently challenging due to the severe scarcity of open-source circuit designs, which are valuable IPs for semiconductor companies. Most available designs are either too trivial or unsuitable for systematic evaluation. Previous work (Yao et al., 2024) relies on contrived designs, failing to address these issues. We handcrafted 36 representative designs that cover diverse implementation types and embody established optimization patterns. Although not at very large scale, RTL-OPT offers sufficient breadth and realism to serve as a valuable resource for advancing LLM-based RTL optimization.

The remainder of this paper is organized as follows: Section 2 provides a systematic analysis of RTL code optimization, including the impact of synthesis, evaluation of existing and new benchmarks, and case studies. Based on the analysis, Section 3 introduces RTL-OPT, a new benchmark on RTL code optimization that addresses our observed challenges. In Section 4, we present the experimental results of different LLMs on RTL-OPT.

Figure 1: The workflow of RTL-OPT for automated benchmarking RTL optimization.

## 2 RETHINKING THE RTL CODE OPTIMIZATION

In this section, we introduce our comprehensive study to inspect both existing benchmark Yao et al. (2024) on RTL code optimization and our own RTL-OPT benchmark under multiple synthesis setups. This study reflects the limitation of overly contrived designs in existing benchmark.

### 2.1 IMPACT OF SYNTHESIS PROCESS ON RTL EVALUATION

According to our study, we point out that the evaluation of RTL optimization (i.e., judging which RTL code leads to better PPA) is not a straightforward task. **One primary reason is that the ultimate design quality also depends on the synthesis process**, which converts the RTL code to the circuit implementation. The synthesis process not only affects the ultimate PPA values, but also the comparison result between a pair of RTL codes. Specifically, differences in *synthesis tools*, *optimization modes*, and *timing constraints* can all significantly affect whether and how structural differences in RTL code are reflected in the final implementation.

**Effect of Synthesis Tool.** Synthesis tools can be broadly categorized into commercial and open-source options, with Synopsys Design Compiler (DC) (des, 2021) and Yosys (Wolf et al., 2013) being the most widely used representatives in each category. DC is an industry-standard tool offering advanced optimization capabilities and robust handling of complex RTL constructs. In contrast, Yosys is an open-source weaker alternative valued in academic research. These tools implement varying optimization strategies and heuristics, so the same RTL may produce significantly different outcomes depending on the chosen tool, directly influencing how design quality is perceived.

**Effect of Compile Mode Selection.** Commercial tools like Synopsys DC support multiple compile modes. For instance, `compile_ultra` applies more aggressive and advanced logic optimizations compared to the basic `compile` mode. Its aggressive optimization by flattening or restructuring logic tends to obscure fine-grained RTL differences.

**Effect of Clock Period Constraints.** The target clock period also shapes synthesis behavior. Tight constraints often lead to aggressive timing-driven optimizations, while relaxed constraints may reduce differentiation between RTL variants. Choosing a realistic and consistent timing target is important for fair and interpretable evaluation of RTL code.

### 2.2 INSPECTION OF EXISTING BENCHMARK

The existing benchmark (Yao et al., 2024) provides multiple pairs of *sub-optimal* and *human-optimized* RTL designs, along with additional RTL code generated by their LLM-based optimization experiments. Surprisingly, our study reveals that: 1) Both the *human-optimized* RTL designs and the *LLM-optimized* RTL designs from (Yao et al., 2024) often fail to outperform their corresponding *sub-optimal* counterparts after synthesis. In many cases, they are essentially the same or even worse, particularly when advanced synthesis options are applied. 2) We observe clearly different impacts on ultimate PPAs between different synthesis tools: commercial tool DC with strong optimization capabilities tend to eliminate the differences between sub-optimal and optimized RTL, while open-source Yosys often exaggerates them. Together, these results suggest that the existing benchmark does not reliably reflect true improvements in RTL code.

| Benchmark | Total | Yosys | | | DC (compile, clk = 1ns) | | | | DC (compile_ultra, clk = 1ns) | | | |
|---|---|---|---|---|---|---|---|---|---|---|---|---|
| | | *same* | *worse* | *better* | *same* | *trade-off* | *worse* | *better* | *same* | *trade-off* | *worse* | *better* |
| Benchmark of (Yao et al., 2024) | 43 | 13 | 6 | 24 | 13 | 7 | 7 | 16 | 21 | 1 | 8 | 13 |
| Paper of (Yao et al., 2024) | 12 | 1 | 0 | 11 | 1 | 4 | 2 | 5 | 4 | 1 | 3 | 4 |
| SymRTLO (Wang et al., 2025) | 13 | 2 | 1 | 10 | 2 | 2 | 1 | 8 | 4 | 1 | 3 | 5 |
| **RTL-OPT** | 36 | 3 | 0 | 33 | 0 | 6 | 0 | 30 | 0 | 1 | 0 | 35 |

Table 1: Comparison between each pair of *sub-optimal* and *human-optimized* designs from (Yao et al., 2024) and RTL-OPT (this work). RTL-OPT shows consistent improvements (35 out of 36 better under `compile_ultra`), matching the expectation that expert-optimized RTL should outperform sub-optimal versions. In contrast, prior benchmarks often show little or no improvement under stronger synthesis settings, indicating limited reliability for benchmarking RTL optimization.

| Benchmark | Total | DC (compile, clk = 0.1ns) | | | | DC (compile_ultra, clk = 0.1ns) | | | | DC (compile_ultra, clk = 1ns) | | | |
|---|---|---|---|---|---|---|---|---|---|---|---|---|---|
| | | *same* | *trade-off* | *worse* | *better* | *same* | *trade-off* | *worse* | *better* | *same* | *trade-off* | *worse* | *better* |
| Benchmark (Yao et al., 2024) | 43 | 13 | 9 | 6 | 15 | 22 | 3 | 7 | 11 | 21 | 1 | 8 | 13 |
| Paper (Yao et al., 2024) | 12 | 1 | 4 | 1 | 6 | 4 | 1 | 1 | 6 | 4 | 1 | 3 | 4 |
| SymRTLO (Wang et al., 2025) | 13 | 2 | 2 | 2 | 7 | 4 | 0 | 3 | 6 | 4 | 1 | 3 | 5 |
| **RTL-OPT** | 36 | 0 | 13 | 0 | 23 | 0 | 12 | 0 | 24 | 0 | 1 | 0 | 35 |

Table 2: Extension of Table 1, evaluating different synthesis setup: clock period as 0.1ns and 1ns.

The evaluation results of existing benchmarks are shown in Table 1, 2 and 3. We carefully inspect and evaluate all 43 pairs of RTL code[1] from the whole benchmark released in (Yao et al., 2024). Specifically, Table 1 compares each pair of *sub-optimal* and *human-optimized* designs, both from the original benchmark. We evaluate whether the *human-optimized* reference is actually better, worse, or the same compared with its *sub-optimal* RTL counterpart after synthesis[2]. In addition, there may exist a "trade-off" result in the PPA comparison, indicating improvement in one PPA metric while degradation in the other. As for Yosys, similar to prior works, we only compare the number of cells. In Table 1, only 13 *human-optimized* RTL (Yao et al., 2024) out of 43 cases are better than their *sub-optimal* counterparts with `compile_ultra`. This number rises to 16 with `compile` and further to 24 with Yosys. Many *human-optimized* RTLs are no better than *sub-optimal* RTL, particularly with advanced synthesis options. It validates that commercial tools can eliminate many contrived inefficiencies, while open-source tools often retain them, highlighting a clear discrepancy.

Table 2 extends our evaluation in Table 1 under different clock constraints, setting a tighter timing target of clock period = 0.1ns. When using the same synthesis modes (i.e., `compile` or `compile_ultra`), using a tighter timing constraint leads to slightly more cases with PPA *trade-offs* and even less actually *better* RTL code.

Table 3 further compares the sub-optimal design with *LLM-optimized* designs directly released by prior work (Yao et al., 2024). In Table 3, for both GPT-4.0 and model proposed by (Yao et al., 2024), only 3 *LLM-optimized* RTL out of 12 cases are actually better than their *sub-optimal* counterparts with `compile_ultra`. The number rises to 5 out of 12 with `compile`. In summary, many *LLM-*

| LLM Solution | Total | Yosys | | | DC (compile, clk = 1ns) | | | | DC (compile_ultra, clk = 1ns) | | | |
|---|---|---|---|---|---|---|---|---|---|---|---|---|
| | | *same* | *worse* | *better* | *same* | *trade-off* | *worse* | *better* | *same* | *trade-off* | *worse* | *better* |
| GPT-4.0 | 13 | 4 | 3 | 6 | 3 | 3 | 2 | 5 | 4 | 4 | 2 | 3 |
| Model (Yao et al., 2024) | 12 | 3 | 1 | 8 | 3 | 4 | 0 | 5 | 3 | 6 | 0 | 3 |

Table 3: Comparison between each pair of *sub-optimal* and *LLM-optimized* designs released from (Yao et al., 2024). Only 14 LLM-optimized designs used in the paper (Yao et al., 2024) are released.

---

[1] As shown in Table 1, paper of (Yao et al., 2024) and SymRTLO (Wang et al., 2025) are different subsets of the Benchmark (Yao et al., 2024). We only successfully synthesized 43 cases out of the 54 pairs of RTL code from the benchmark (Yao et al., 2024). For the others, we synthesized 12 out of 14 pairs and 13 out of 16 pairs, respectively. These synthesis failures in the original benchmarks are mainly caused by Verilog syntax errors.

[2] The details of the synthesis process, tools, and PPA metrics used for evaluating are provided in Section 3.3.

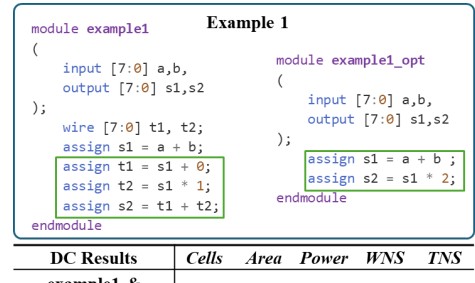

Figure 2: Overly contrived suboptimal and optimized RTL code in existing benchmark (Yao et al., 2024). Both codes have the same PPA after synthesis with the commercial tool DC.

*optimized* RTLs turn out to be no better than *sub-optimal* RTL, especially with advanced synthesis options. It indicates that under strict synthesis flows, the benchmark shows limited effectiveness. We provide detailed data in Appendix G.2.

## 2.3 SAME INSPECTION OF OUR BENCHMARK

In Table 1 and 2, we also evaluate our proposed benchmark, RTL-OPT, using the same setup. According to Table 1, 35 out of 36 *human-optimized* RTL codes in RTL-OPT are better when `compile_ultra` is adopted, and 33 out of 36 for Yosys. In Table 2, with a tight timing constraint, 23 cases remain better while 13 result in PPA trade-offs, with no cases achieving the same ultimate PPA. Compared to the previous benchmark (Yao et al., 2024), RTL-OPT shows significant improvements in evaluating RTL designs.

This clear validation of RTL-OPT's benchmark quality arises from its design philosophy: it provides genuinely *sub-optimal* RTL implementations with meaningful room for improvement, rather than contrived inefficiencies that synthesis tools can remove. The detailed comparisons between the suboptimal and optimized RTL designs in RTL-OPT are in Appendix C.

## 2.4 CASE STUDY OF EXISTING BENCHMARK

To further understand how such contrived RTL designs are constructed, we take a closer look at (Yao et al., 2024) benchmark. *Example 1 & 2* in Figure 2 highlight a common flaw in (Yao et al., 2024) designs: its RTL pairs are based on unrealistic transformations that do not address true optimization challenges in hardware design. These contrived sub-optimal examples exhibit unnecessary inefficiencies, such as redundant computations and superfluous arithmetic operations, which are unlikely to occur in practice. Synthesis tools can easily optimize these contrived patterns, leading to evaluations that may overstate the effectiveness of LLMs in improving RTL quality.

In Example 1 (Yao et al., 2024), the optimized version (`example1_opt`) implements the logic directly by computing `s1 = a + b` and then deriving the output as `s2 = s1 * 2`. In contrast, the suboptimal version (`example1`) introduces contrived and unnecessary steps: it first computes `s1 = a + b`, then redundantly adds 0 and multiplies by 1 to produce intermediate wires `t1` and `t2`, before summing them into `s2`. Such constructions are unnatural and would rarely appear in practical RTL coding, making the benchmark example unrealistic.

In Example 2 (Yao et al., 2024), the optimized version (`example2_opt`) simply applies constant folding to collapse the entire computation into one step: `z = c * 67`. By contrast, the suboptimal version (`example2`) artificially expands this trivial logic into a sequence of assignments, first setting `a = 2`, then computing `b = (a * 32) + 3`, and finally multiplying `b` by input `c` to obtain `z`. These contrived constructions of sub-optimal cases result in an unrealistic benchmark.

In summary, these examples highlight that sub-optimal cases in (Yao et al., 2024) rely on contrived inefficiencies rather than realistic IC design challenges. Such cases are unrepresentative of practical RTL design and can be trivially optimized by synthesis tools, limiting the benchmark's ability to assess LLMs on RTL optimization.

## 3 RTL-OPT BENCHMARK

In this section, we present RTL-OPT, a benchmark designed for evaluating RTL code optimization with LLMs. RTL-OPT provides realistic suboptimal and optimized RTL pairs handcrafted by experts, ensuring genuine inefficiencies and meaningful golden references. Covering diverse design types and evaluated with both commercial and open-source tools, it offers a robust and practical resource for advancing RTL optimization research.

### 3.1 BENCHMARK DESCRIPTION

The RTL-OPT consists of 36 RTL design optimization tasks. Each task provides a pair of RTL codes: a suboptimal version and a corresponding designer-optimized version, implementing the same functionality. All designs are manually written by hardware engineers to reflect realistic coding styles and optimization practices, with the optimized RTL serving as the golden reference for human-optimized PPA quality. The suboptimal RTL is not arbitrarily degraded; it represents a valid, functionally correct design that omits specific optimization opportunities. This setup creates meaningful optimization gaps and practical scenarios encountered in the semiconductor industry.

The 36 provided design tasks cover a variety of design types, including arithmetic units, control logic, finite state machines (FSMs), and pipelined datapaths. These designs vary in size and complexity, with logic area ranging from **14** to **20K** cells and synthesized area ranging from **15** to **19K** $\mu m^2$. This diversity ensures that the benchmark is representative of practical RTL design tasks.

RTL-OPT is fully open-sourced and provides the following artifacts to support benchmarking the RTL optimization capabilities of LLMs: (1) 36 carefully designed RTL code pairs; (2) Corresponding synthesized netlists from commercial synthesis tools; (3) Detailed PPA reports from electronic design automation (EDA) tools for both suboptimal and optimized designs; (4) A complete toolchain flow, including scripts for synthesis, simulation, and functional verification, which can also verify the correctness of the rewritten code by LLMs.

Table 4 shows the evaluated PPA of all 36 pairs of sub-optimal and optimized designs from RTL-OPT, using both **DC** and **Yosys**. Due to its size, this table is now in Appendix C.1. Detailed explanation of our evaluation methodology, synthesis process, and the PPA metrics are provided in Section 3.3.

### 3.2 RTL-OPT ANALYSIS: OPTIMIZATION PATTERNS

The *optimization patterns*, which provide optimization opportunities, are derived from proven industry-standard RTL coding practices that have a direct impact on the quality of logic synthesis. These patterns represent how specific RTL-level modifications ultimately affect downstream synthesis outcomes. The key optimization pattern types in the RTL-OPT benchmark are summarized as follows:

- **Bit-width Optimization:** Reducing register and wire widths where full precision is not necessary, optimizing both area and power consumption.

- **Precomputation & LUT Conversion:** Replacing runtime arithmetic operations with precomputed lookup tables to eliminate complex logic units.

- **Operator Strength Reduction:** Substituting high-cost operators with simpler equivalents through bit manipulation.

- **Control Simplification:** Flattening nested finite state machines (FSMs) or reducing unnecessary states, streamlining control logic, and improving both area and timing.

- **Resource Sharing:** Consolidating duplicate logic across different cycles to maximize hardware resource efficiency.

- **State Encoding Optimization:** Selecting optimal state encoding schemes (One-hot, Gray, Binary) based on state count to balance power, area, and timing.

By integrating these optimization patterns across a diverse range of RTL designs, RTL-OPT generates its realistic yet challenging benchmark for LLM-assisted RTL code optimization: enhancing PPA metrics of optimized code while maintaining functional correctness.

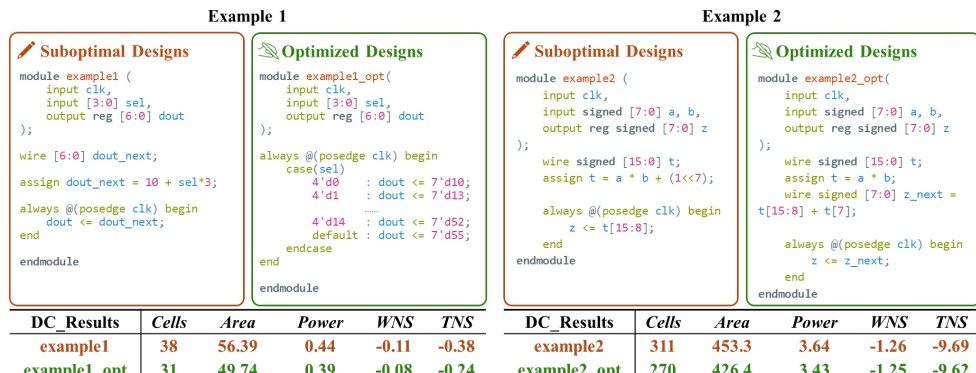

Figure 3: Comparison of suboptimal and optimized RTL code examples in RTL-OPT.

To illustrate these optimization patterns, we provide *two code examples* from the RTL-OPT in Figure 3. These examples compare suboptimal and optimized RTL implementations within a specific pattern category, accompanied by discussions on the structural changes and the quantitative improvements observed in downstream PPA metrics.

**Example 1:** This example in Figure 3 (left) demonstrates the optimization pattern of **precomputation & LUT conversion**, where real-time arithmetic operations are replaced with precomputed values. In the suboptimal design, the output is dynamically calculated using the `sel` input, requiring multiplication at each clock cycle. The optimized design replaces this operation with a `case` statement that directly assigns precomputed values based on the selection. This optimization results in a 14% reduction in area and a 12% decrease in power consumption by eliminating arithmetic operations and reducing signal toggling.

**Example 2:** This example in Figure 3 (right) demonstrates **bit-width optimization**, where the physical implementation of an arithmetic operation is restructured to minimize resource usage. In the suboptimal design, a 16-bit multiplication is followed by an addition of the least significant bit, resulting in a larger bit-width for intermediate signals. The optimized design reduces the bit-width by truncating the multiplication result to an 8-bit value directly and simplifying the addition operation. This restructuring achieves a 7% reduction in area and a 6% decrease in power consumption by reducing the width of intermediate signals and operations.

## 3.3 EVALUATION METHODOLOGY AND TOOLS

RTL-OPT provides a complete evaluation flow to assess LLMs' optimization capabilities by measuring the PPA of synthesized RTL code. This is achieved through a combination of synthesis, functionality verification, and PPA evaluation, all performed using industry-standard EDA tools.

### 3.3.1 SYNTHESIS PROCESS

The logic synthesis process converts the initial RTL code into gate-level netlists, based on which the PPA metrics can be quantitatively evaluated. In this work, we mainly employ DC (des, 2021) for the synthesis of the RTL-OPT benchmark, given its established effectiveness in industrial design flows. DC demonstrates superior capabilities in identifying inefficient RTL constructs and optimizing them into more efficient circuit implementations, thereby minimizing sensitivity to the initial code quality.

When benchmark quality is insufficient, DC tends to synthesize both the suboptimal and optimized RTL codes into functionally equivalent gate-level netlists, resulting in identical PPA outcomes. This behavior reflects the limited optimization opportunities offered by low-quality benchmarks. Conversely, open-source synthesis tools such as Yosys (Wolf et al., 2013), which provide less aggressive optimization, may still produce differing PPA results for such code pairs, potentially overstating the effectiveness of certain code transformations. For completeness, we also provide synthesis results obtained using Yosys to support broader comparative analyses.

The synthesis process also involves the use of a technology library, or cell library, which is a collection of pre-characterized standard cells such as logic gates, flip-flops, and other fundamental components.

These cells are designed to meet specific PPA constraints. While we use Nangate45 (Si2, 2018) in our evaluation, other libraries could also be used, though they typically require a license. The choice of library significantly impacts the RTL optimization process, as it defines the available cells and their performance characteristics, ultimately influencing the design's efficiency.

### 3.3.2 FUNCTIONAL EQUIVALANCE VERIFICATION

After successful synthesis, the RTL code is ensured to be free of syntax errors, as it can be correctly transformed into a gate-level netlist. Following synthesis, functional verification is essential to ensure that the optimization steps have not introduced errors. This verification is primarily conducted using **Synopsys Formality** (fom, 2023), which performs functional equivalence checking by rigorously comparing the LLM-optimized RTL against the golden reference to ensure behavioral consistency. However, for optimizations involving timing adjustments, such as pipelining, additional dynamic verification is required. This is performed using **Synopsys VCS** (vcs, 2021), which employs comprehensive testbenches to validate the design's behavior under various operating conditions.

This combined approach ensures both logical equivalence and operational reliability of the optimized design. A design is considered functionally valid only if it passes both formal equivalence checking and dynamic verification for timing-critical optimizations.

### 3.3.3 PPA METRICS AND TRADE-OFFS IN OPTIMIZATION

To evaluate the quality of the synthesized designs, we analyze them from three aspects: **Power**, **Performance**, and **Area**:

**Power:** The total power consumption of the synthesized design, characterized by the fundamental equation: $P_{\text{dynamic}} = \alpha C V^2 f$, where $\alpha$ is the switching activity, $C$ is the capacitance, $V$ is the supply voltage, and $f$ is the clock frequency.

**Performance:** Evaluated through two key timing metrics: *Worst Negative Slack (WNS)*, which represents the largest single timing violation in the design, and *Total Negative Slack (TNS)*, the sum of all timing violations across failing paths.

**Area**: Characterized by two complementary measures: *Silicon area* (in $\mu m^2$), which indicates the physical implementation footprint, and *Cell count*, the total number of standard cells in the design, providing a basic area estimation that does not account for cell types, placement, or routing overhead.

Trade-offs widely exist in these PPA metrics. For instance, optimizing for power may increase area, while minimizing area could compromise power efficiency. A key challenge in RTL optimization is managing these competing goals to achieve an optimal balance based on design constraints.

## 4 EXPERIMENTS

This section presents the experiments conducted to evaluate the optimization capabilities of different LLMs on the RTL-OPT benchmark. We compare the performance of several LLMs, including **GPT-4o-mini**, **Gemini-2.5**, **Deepseek V3**, and **Deepseek R1**, in optimizing RTL code. The focus of the experiments is on assessing the optimization in terms of PPA metrics, as well as the functional correctness of the optimized designs. The results show that the two Deepseek models demonstrate stronger optimization ability than the other evaluated LLMs. Detailed tables summarizing PPA performance and functional correctness (Table 6) are included in Appendix D.1.

### 4.1 SUMMARY OF BENCHMARKING RESULTS

Figure 4 shows a summary of benchmarking results of the four evaluated LLMs. It reveals the syntax correctness, functionality correctness, and post-optimization PPA quality performance of the various LLMs. The overall results highlight that: ❶ There is still significant room for LLM to improve in RTL optimization compared to human designers. ❷ Our benchmark is designed to be realistic, providing a set of challenging tasks that reflect the complexities encountered in real-world hardware design.

Notably, the overall performance of all LLMs is not very good, reflecting the challenges in our RTL-OPT benchmark. Many LLMs have over 10 optimized cases failed to maintain functionality cor-

rectness. Deepseek R1 can successfully optimize about 15 sub-optimal designs, and can outperform our human designers' solution for around 5 designs.

When comparing these 4 LLMs, Deepseek R1 generally outperforms the other models in terms of PPA. However, Deepseek R1 also exhibits a higher rate of functional discrepancies compared to the other models. In contrast, models such as GPT-4o-mini and Gemini-2.5, while maintaining high syntax correctness, achieved fewer improvements in PPA. It may imply that their optimization strategies are either more conservative or lack effective optimizations.

## 4.2 DETAILED BENCHMARKING RESULTS

One observation from the evaluation results in Figure 4 is the trade-off between optimization and functionality. While Deepseek R1 showed the most significant improvements in PPA, it was also the most prone to introducing functional errors. In contrast, GPT-4o-mini and Gemini-2.5 exhibited a more balanced approach, optimizing PPA while maintaining syntax correctness. This indicates that Deepseek R1's aggressive optimization, though effective, tends to increase error, especially in designs with complex timing or control logic. Conversely, GPT-4o-mini and Gemini-2.5, while less aggressive, maintained functional correctness and achieved meaningful PPA improvements. We provide more detailed LLM evaluation results in Appendix E.

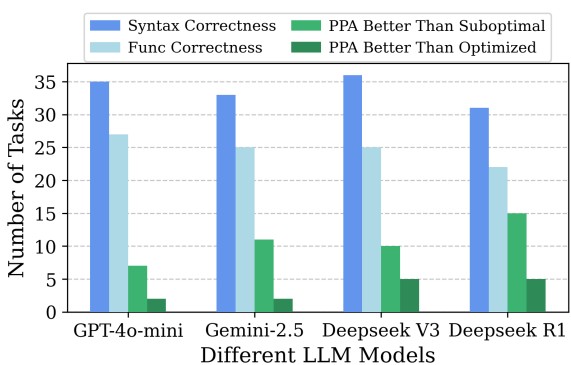

Figure 4: Comparison of optimization performance across different LLMs.

For the experimental results, we summarize the performance of four LLMs across syntax correctness, functional correctness, and PPA improvement. 1) **GPT-4o-mini** achieves good correctness, with a syntax correctness rate of 97.2% and functional correctness of 75%. Though only 19.4% of its generated code achieving better PPA than the suboptimal version. 2) Similarly, **Gemini-2.5** exhibited the same trend as GPT-4o-mini: relatively high functional correctness but low performance in PPA optimization. 3) For **Deepseek V3**, it gets the highest syntax correctness of 100%, and the same functional correctness of 69.4% with Gemini-2.5. It achieved a balanced performance across all metrics. 4)In contrast, **Deepseek R1**, with a syntax correctness rate of 86.1% and functional correctness of 61.1%, produced 41.7% of the code with better PPA than the suboptimal version, and 13.9% better than designer solutions, despite its lower functional correctness.

Beyond quantitative results, we also randomly inspected 40 cases where LLM-optimized designs passed syntax checks but failed functional verification. We observed three main failure modes: **control logic inconsistencies** (e.g., incorrect Boolean conditions in comparators), **overly aggressive pipelining** (e.g., violating latency requirements in FSMs), and **improper resource sharing** (e.g., stale data due to register reuse). These results highlight that LLM errors often stem from subtle design semantics rather than surface-level syntax issues. We provide the details in Appendix F.

## 5 CONCLUSIONS AND LIMITATIONS

In this paper, we introduce RTL-OPT, a benchmark for hardware RTL code optimization aimed at enhancing IC design quality. RTL-OPT includes 36 handcrafted digital IC designs, each with suboptimal and optimized RTL code, enabling the assessment of LLM-generated RTL. An integrated evaluation framework verifies functional correctness and quantifies PPA improvements, providing a standardized method for evaluating generative AI models in hardware design. RTL-OPT has significant potential to influence AI-assisted IC design by offering valuable insights and fostering advancements. As for the limitations of RTL-OPT, it relies entirely on expert-written, manually optimized RTL code, limiting the dataset's scale. Expanding to a larger dataset requires advances in automated optimization or synthetic generation of high-quality RTL, which remains challenging.

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

## A  LLM USAGE

Large Language Models (LLMs) were used to aid in the writing and polishing of the manuscript. Specifically, we used an LLM to assist in refining the language, improving readability, and ensuring clarity in various sections of the paper. The model helped with tasks such as sentence rephrasing, grammar checking, and enhancing the overall flow of the text.

It is important to note that the LLM was not involved in the ideation, research methodology, or experimental design. All research concepts, ideas, and analyses were developed and conducted by the authors. The contributions of the LLM were solely focused on improving the linguistic quality of the paper, with no involvement in the scientific content or data analysis.

The authors take full responsibility for the content of the manuscript, including any text generated or polished by the LLM. We have ensured that the LLM-generated text adheres to ethical guidelines and does not contribute to plagiarism or scientific misconduct.

## B  ETHICS STATEMENT

This work adheres to the ICLR Code of Ethics. In this study, no human subjects or animal experimentation was involved. All datasets used, including RTL-OPT, were sourced in compliance with relevant usage guidelines, ensuring no violation of privacy. We have taken care to avoid any biases or discriminatory outcomes in our research process. No personally identifiable information was used, and no experiments were conducted that could raise privacy or security concerns. We are committed to maintaining transparency and integrity throughout the research process.

## C    FULL PPA QUALITY COMPARISON OF RTL-OPT

Due to space limitations, we move the full PPA evaluation results to the appendix. This section provides the complete comparison between the suboptimal and optimized RTL designs in RTL-OPT. Both Synopsys Design Compiler and Yosys are used for synthesis and evaluation. As PPA inherently involves trade-offs, smaller values indicate better design quality. These results complement the main text by presenting full numerical evidence under different synthesis constraints.

### C.1    RESULTS UNDER DC COMPILE_ULTRA, 1 NS

Table 4 reports the full PPA metrics of all 36 RTL-OPT designs when synthesized using Synopsys DC with a relaxed 1ns clock period on the "compile ultra" setting. Both suboptimal and optimized RTL codes are evaluated, enabling direct comparison. This setting highlights how expert-optimized RTL consistently achieves superior power, performance, and area outcomes compared to the suboptimal versions, demonstrating the reliability of RTL-OPT for benchmarking.

| | Design List | Provided Suboptimal Designs | | | | | | | Provided Optimized Designs | | | | | | |
| | | DC Results(*compile_ultra, 1ns*) | | | | | Yosys Results | | DC Results(*compile_ultra, 1ns*) | | | | | Yosys Results | |
| | | Cells | Area | *Power* | *WNS* | *TNS* | Cells | Area | Cells | Area | *Power* | *WNS* | *TNS* | Cells | Area |
|---|---|---|---|---|---|---|---|---|---|---|---|---|---|---|---|
| 1 | adder | 397 | 510.4 | 370.4 | 0.0 | 0.0 | 323 | 456.2 | 372 | 477.2 | 351.3 | 0.0 | 0.0 | 266 | 413.9 |
| 2 | adder_select | 450 | 439.2 | 264.3 | 0.0 | 0.0 | 363 | 428.3 | 306 | 225.7 | 0.23 | 0.0 | 0.0 | 241 | 318.4 |
| 3 | alu_64bit | 1683 | 1645 | 887.8 | 0.0 | 0.0 | 1411 | 1554 | 1680 | 1557 | 677.1 | 0.0 | 0.0 | 1094 | 1307 |
| 4 | alu_8bit | 146 | 180 | 75.58 | 0.0 | 0.0 | 152 | 163.1 | 115 | 138 | 68.43 | 0.0 | 0.0 | 130 | 158.8 |
| 5 | calculation | 888 | 1044 | 0.78 | -0.61 | -5.66 | 774 | 900.4 | 755 | 860 | 0.62 | -0.60 | -4.83 | 567 | 661.8 |
| 6 | comparator | 63 | 56.39 | 0.03 | 0.0 | 0.0 | 40 | 40.70 | 45 | 41.76 | 0.02 | 0.0 | 0.0 | 40 | 40.70 |
| 7 | comparator_16bit | 102 | 88.31 | 44.57 | 0.0 | 0.0 | 64 | 62.24 | 116 | 105.3 | 38.83 | 0.0 | 0.0 | 104 | 105.1 |
| 8 | comparator_2bit | 10 | 9.58 | 4.51 | 0.0 | 0.0 | 11 | 9.31 | 10 | 8.78 | 3.82 | 0.0 | 0.0 | 10 | 9.31 |
| 9 | comparator_4bit | 23 | 21.28 | 10.86 | 0.0 | 0.0 | 22 | 21.55 | 21 | 18.35 | 9.25 | 0.0 | 0.0 | 18 | 17.29 |
| 10 | comparator_8bit | 46 | 44.42 | 22.8 | 0.0 | 0.0 | 45 | 48.15 | 47 | 41.23 | 20.29 | 0.0 | 0.0 | 33 | 30.86 |
| 11 | decoder_6bit | 97 | 76.61 | 22.74 | 0.0 | 0.0 | 93 | 99.22 | 87 | 71.55 | 19.54 | 0.0 | 0.0 | 91 | 78.20 |
| 12 | decoder_8bit | 317 | 257.2 | 50.67 | 0.0 | 0.0 | 390 | 410.7 | 312 | 246.8 | 50.28 | 0.0 | 0.0 | 300 | 250 |
| 13 | divider_16bit | 3471 | 3428 | 5088 | -3.43 | -75.6 | 1264 | 1412 | 1565 | 1560 | 2091 | -3.83 | -72.1 | 662 | 760.2 |
| 14 | divider_32bit | 14400 | 14296 | 23796 | -10.69 | -486 | - | - | 6724 | 6703.20 | 11335 | -10.19 | -468 | 2419 | 2941 |
| 15 | divider_4bit | 39 | 40.43 | 21.58 | 0.0 | 0.0 | 52 | 59.05 | 33 | 33.78 | 15.9 | 0.0 | 0.0 | 39 | 40.96 |
| 16 | divider_8bit | 571 | 575.4 | 681.8 | -0.50 | -4.57 | 302 | 339.7 | 322 | 330.37 | 275.2 | -0.39 | -2.43 | 171 | 189.7 |
| 17 | fsm | 89 | 128.7 | 92.56 | 0.0 | 0.0 | 85 | 138.6 | 73 | 92.04 | 51.44 | 0.0 | 0.0 | 70 | 97.09 |
| 18 | fsm_encode | 242 | 387.0 | 320.6 | 0.0 | 0.0 | 179 | 352.7 | 155 | 305.63 | 276.97 | 0.0 | 0.0 | 170 | 319.2 |
| 19 | gray | 48 | 67.30 | 85.32 | 0.0 | 0.0 | 63 | 85.92 | 51 | 69.69 | 66.78 | 0.0 | 0.0 | 67 | 88.84 |
| 20 | mac | 410 | 717.9 | 868.2 | 0.0 | 0.0 | 532 | 831.5 | 319 | 548.23 | 735.25 | 0.0 | 0.0 | 529 | 707 |
| 21 | mul | 315 | 378.5 | 421.7 | -0.05 | -0.13 | 399 | 485.5 | 315 | 378.5 | 421.7 | -0.05 | -0.13 | 397 | 476.4 |
| 22 | mul_sub | 234 | 338.09 | 262.35 | 0.0 | 0.0 | 299 | 352.9 | 233 | 337.02 | 256.53 | 0.0 | 0.0 | 289 | 332.5 |
| 23 | mux | 25 | 31.92 | 10.38 | 0.0 | 0.0 | 8 | 8.51 | 25 | 21.81 | 8.38 | 0.0 | 0.0 | 34 | 42.29 |
| 24 | mux_encode | 125 | 140.7 | 0.43 | 0.0 | 0.0 | - | - | 34 | 36.18 | 0.13 | 0.0 | 0.0 | - | - |
| 25 | saturating_add | 24 | 69.43 | 67.53 | 0.0 | 0.0 | 58 | 97.36 | 18 | 67.56 | 65.77 | 0.0 | 0.0 | 42 | 78.47 |
| 26 | selector | 18 | 39.37 | 36.06 | 0.0 | 0.0 | 17 | 42.56 | 18 | 38.04 | 35.32 | 0.0 | 0.0 | 15 | 37.24 |
| 27 | sub_16bit | 132 | 136.2 | 77.92 | 0.0 | 0.0 | 93 | 96.56 | 124 | 131.9 | 75.46 | 0.0 | 0.0 | 92 | 98.95 |
| 28 | sub_32bit | 270 | 251.9 | 148.9 | 0.0 | 0.0 | 189 | 191.5 | 265 | 244.7 | 142.52 | 0.0 | 0.0 | 188 | 200 |
| 29 | sub_4bit | 12 | 18.35 | 9.29 | 0.0 | 0.0 | 21 | 22.34 | 10 | 17.82 | 9.29 | 0.0 | 0.0 | 20 | 22.08 |
| 30 | sub_8bit | 25 | 41.76 | 24.96 | 0.0 | 0.0 | 45 | 47.88 | 20 | 36.97 | 19.26 | 0.0 | 0.0 | 46 | 48.15 |
| 31 | add_sub | 164 | 183.27 | 118.4 | 0.0 | 0.0 | 155 | 164.4 | 124 | 130.34 | 101.7 | 0.0 | 0.0 | 179 | 202.7 |
| 32 | addr_calcu | 78 | 131.40 | 96.07 | -0.03 | -0.06 | 197 | 222.9 | 82 | 125.55 | 90.27 | -0.01 | -0.01 | 101 | 124.7 |
| 33 | mult_if | 10 | 10.91 | 3.53 | 0.0 | 0.0 | 12 | 10.9 | 11 | 10.11 | 4.13 | 0.0 | 0.0 | 12 | 10.91 |
| 34 | mux_large | 81 | 97.62 | 48.27 | 0.0 | 0.0 | 65 | 90.17 | 81 | 96.82 | 40.84 | 0.0 | 0.0 | 112 | 120.5 |
| 35 | register | 3731 | 8745 | 7712 | 0.0 | 0.0 | 4500 | 9735 | 3720 | 8744 | 7708 | 0.0 | 0.0 | 4507 | 9668 |
| 36 | ticket_machine | 36 | 58.52 | 60.31 | 0.0 | 0.0 | 29 | 47.88 | 22 | 32.45 | 36.03 | 0 | 0 | 29 | 47.88 |

Table 4: The PPA quality comparison of RTL-OPT-provided suboptimal vs. optimized designs(compile_ultra, 1ns). Using both commercial DC and open-source Yosys for the RTL design synthesis and PPA evaluations. Trade-offs are common in these PPA metrics and smaller values indicate better performance.

## C.2 RESULTS UNDER DC COMPILE, 0.1 NS

Table 5 presents the full PPA metrics for the same 36 designs under a more aggressive 0.1ns clock period constraint. Compared to the 1ns setting, these results illustrate sharper trade-offs among PPA metrics, where aggressive timing optimization can sometimes increase power or area. Nevertheless, the optimized RTL consistently outperforms the suboptimal RTL, reaffirming that RTL-OPT reflects realistic and meaningful optimization challenges.

| | Design List | Provided Suboptimal Designs | | | | | | | | Provided Optimized Designs | | | | | | | |
| | | DC Results (*compile, 0.1ns*) | | | | | Yosys Results | | | DC Results (*compile, 0.1ns*) | | | | | Yosys Results | | |
| | | Cells | Area | *Power* | *WNS* | *TNS* | Wires | Cells | Area | Cells | Area | *Power* | *WNS* | *TNS* | Wires | Cells | Area |
|---|---|---|---|---|---|---|---|---|---|---|---|---|---|---|---|---|---|
| 1 | adder | 626 | 688.4 | 5.1 | 0.57 | 16.0 | 501 | 323 | 456.2 | 531 | 639.7 | 4.26 | 0.46 | 13.2 | 440 | 266 | 413.9 |
| 2 | adder_select | 813 | 812.1 | 4.85 | 0.39 | 10.8 | 724 | 363 | 428.3 | 514 | 522.6 | 3.49 | 0.42 | 12.0 | 490 | 241 | 318.4 |
| 3 | alu_64bit | 3248 | 3028 | 16.95 | 0.51 | 29.7 | 2358 | 1411 | 1554 | 1706 | 1748.9 | 8.58 | 0.74 | 42.7 | 1791 | 1094 | 1307 |
| 4 | alu_8bit | 402 | 370 | 1.89 | 0.26 | 1.91 | 257 | 152 | 163.1 | 244 | 245.8 | 1.15 | 0.40 | 2.92 | 229 | 130 | 158.8 |
| 5 | calculation | 670 | 997.5 | 6.59 | 3.22 | 42.6 | 1210 | 774 | 900.4 | 534 | 761.8 | 5.25 | 3.24 | 44.3 | 927 | 567 | 661.8 |
| 6 | comparator | 109 | 98.15 | 0.43 | 0.19 | 0.19 | 111 | 40 | 40.70 | 90 | 80.60 | 0.34 | 0.21 | 0.21 | 108 | 40 | 40.70 |
| 7 | comparator_16bit | 296 | 244.4 | 1.06 | 0.18 | 0.52 | 141 | 64 | 62.24 | 151 | 169.2 | 0.75 | 0.19 | 0.52 | 357 | 104 | 105.1 |
| 8 | comparator_2bit | 14 | 14.36 | 0.06 | 0.02 | 0.05 | 40 | 11 | 9.31 | 16 | 13.03 | 0.05 | 0.01 | 0.02 | 28 | 10 | 9.31 |
| 9 | comparator_4bit | 25 | 25.80 | 0.11 | 0.06 | 0.15 | 83 | 22 | 21.55 | 29 | 23.14 | 0.09 | 0.10 | 0.22 | 45 | 18 | 17.29 |
| 10 | comparator_8bit | 73 | 71.29 | 0.31 | 0.12 | 0.33 | 170 | 45 | 48.15 | 81 | 67.30 | 0.28 | 0.12 | 0.34 | 77 | 33 | 30.86 |
| 11 | decoder_6bit | 204 | 208 | 0.37 | 0.18 | 9.22 | 106 | 93 | 99.22 | 132 | 106.7 | 0.32 | 0.04 | 2.53 | 195 | 91 | 78.20 |
| 12 | decoder_8bit | 781 | 855.9 | 1.28 | 0.27 | 58.8 | 407 | 390 | 410.7 | 435 | 373.2 | 0.83 | 0.09 | 20.8 | 617 | 300 | 250 |
| 13 | divider_16bit | 5037 | 5045 | 66.3 | 3.88 | 89.4 | 2385 | 1264 | 1412 | 2543 | 2426 | 26.71 | 4.10 | 90.4 | 2116 | 662 | 760.2 |
| 14 | divider_32bit | 21348 | 19849 | 275.51 | 12.26 | 586 | - | - | - | 16434 | 16053 | 164 | 13.68 | 875 | 8341 | 2419 | 2941 |
| 15 | divider_4bit | 85 | 84.85 | 0.48 | 0.32 | 1.29 | 112 | 52 | 59.05 | 64 | 56.13 | 0.23 | 0.17 | 0.59 | 133 | 39 | 40.96 |
| 16 | divider_8bit | 744 | 731.8 | 6.94 | 1.21 | 13.5 | 539 | 302 | 339.7 | 557 | 535.5 | 3.67 | 1.10 | 10.8 | 527 | 171 | 189.7 |
| 17 | fsm | 149 | 192.9 | 1.06 | 0.28 | 4.40 | 209 | 85 | 138.6 | 106 | 129.8 | 0.59 | 0.24 | 2.70 | 157 | 70 | 97.09 |
| 18 | fsm_encode | 353 | 488.9 | 3.6 | 0.41 | 10.8 | 380 | 179 | 352.7 | 334 | 426.7 | 3.07 | 0.39 | 8.11 | 342 | 170 | 319.2 |
| 19 | gray | 100 | 109.1 | 0.92 | 0.28 | 2.04 | 142 | 63 | 85.92 | 81 | 94.70 | 1.04 | 0.23 | 2.34 | 145 | 67 | 88.84 |
| 20 | mac | 563 | 880.7 | 8.46 | 1.23 | 20.9 | 737 | 532 | 831.5 | 477 | 701.4 | 7.26 | 1.38 | 16.2 | 699 | 529 | 707 |
| 21 | mul | 311 | 453 | 3.64 | 1.26 | 9.69 | 498 | 399 | 485.5 | 270 | 426.4 | 3.43 | 1.25 | 9.62 | 494 | 397 | 476.4 |
| 22 | mul_sub | 634 | 623.24 | 4.17 | 0.75 | 8.08 | 494 | 299 | 352.9 | 614 | 603.8 | 4.12 | 0.71 | 8.11 | 480 | 289 | 332.5 |
| 23 | mux | 40 | 38.30 | 0.11 | 0.01 | 0.07 | 27 | 8 | 8.51 | 25 | 31.92 | 0.1 | 0.02 | 0.14 | 68 | 34 | 42.29 |
| 24 | mux_encode | 125 | 140.71 | 0.43 | 0.07 | 0.59 | - | - | - | 34 | 36.18 | 0.13 | 0.08 | 0.58 | - | - | - |
| 25 | saturating_add | 159 | 176.6 | 1.31 | 0.27 | 2.25 | 122 | 58 | 97.36 | 127 | 140.9 | 1.08 | 0.36 | 2.86 | 100 | 42 | 78.47 |
| 26 | selector | 38 | 56.4 | 0.44 | 0.11 | 0.38 | 42 | 17 | 42.56 | 31 | 49.74 | 0.39 | 0.08 | 0.24 | 29 | 15 | 37.24 |
| 27 | sub_16bit | 234 | 223.2 | 1.31 | 0.27 | 3.48 | 220 | 93 | 96.56 | 302 | 263.1 | 1.39 | 0.30 | 3.94 | 181 | 92 | 98.95 |
| 28 | sub_32bit | 542 | 502.2 | 2.94 | 0.31 | 8.49 | 444 | 189 | 191.5 | 518 | 452.2 | 2.51 | 0.36 | 9.22 | 370 | 188 | 200 |
| 29 | sub_4bit | 37 | 36.71 | 0.17 | 0.10 | 0.35 | 52 | 21 | 22.34 | 31 | 29.79 | 0.15 | 0.13 | 0.40 | 43 | 20 | 22.08 |
| 30 | sub_8bit | 135 | 122.6 | 0.65 | 0.15 | 0.95 | 108 | 45 | 47.88 | 129 | 111.9 | 0.57 | 0.26 | 1.47 | 90 | 46 | 48.15 |
| 31 | add_sub | 496 | 444.2 | 2.57 | 0.31 | 4.20 | 277 | 155 | 164.4 | 387 | 356.9 | 2.13 | 0.37 | 4.87 | 323 | 179 | 202.7 |
| 32 | addr_calcu | 405 | 388.1 | 3.07 | 0.62 | 8.56 | 340 | 197 | 222.9 | 229 | 214.4 | 1.8 | 0.60 | 8.47 | 192 | 101 | 124.7 |
| 33 | mult_if | 17 | 15.96 | 0.052 | 0.05 | 0.05 | 39 | 12 | 10.9 | 15 | 14.90 | 0.046 | 0.09 | 0.09 | 34 | 12 | 10.91 |
| 34 | mux_large | 296 | 273.45 | 0.84 | 0.11 | 0.85 | 210 | 65 | 90.17 | 164 | 176.62 | 0.74 | 0.13 | 1.01 | 270 | 112 | 120.5 |
| 35 | register | 5003 | 9780 | 79.63 | 0.40 | 245 | 8096 | 4500 | 9735 | 4481 | 9583 | 77.87 | 0.37 | 244 | 8140 | 4507 | 9668 |
| 36 | ticket_machine | 53 | 74.48 | 0.77 | 0.22 | 1.79 | 73 | 29 | 47.88 | 48 | 51.34 | 0.45 | 0.17 | 0.83 | 73 | 29 | 47.88 |

Table 5: The PPA quality comparison of RTL-OPT-provided suboptimal vs. optimized designs (compile, 0.1ns). Using both commercial DC and open-source Yosys for the RTL design synthesis and PPA evaluations. Trade-offs are common in these PPA metrics and smaller values indicate better performance.

# D PPA AND FUNCTIONAL CORRECTNESS OF LLM-OPTIMIZED DESIGNS

Due to space limitations, the detailed experimental results are moved to the appendix. They report PPA quality and functional correctness for all designs optimized by **GPT-4o-mini**, **Gemini-2.5**, **Deepseek V3**, and **Deepseek R1**, using the RTL-OPT benchmark.

Table 6 and 7 summarize the evaluated PPA performance of each LLM-optimized design and compare it with the provided suboptimal RTL and optimized RTL (golden reference). Green cells indicate that the PPA is better than the suboptimal RTL, and **bold green cells** indicate that the PPA surpasses the optimized RTL (golden reference). The table also shows the functional correctness after

verification (Func column), with ✔ and ✗ representing the verification results. ✗✗ indicates that the corresponding design contains syntax errors and fails to pass DC synthesis.

## D.1 DC COMPILE_ULTRA, 1 NS)

Table 6 shows the PPA quality (DC *compile_ultra, 1ns*) and functional correctness for all LLM-optimized designs. The specific analysis of these results is presented in Section 4.

| Category | GPT-4o-mini | | | | | | Gemini-2.5 | | | | | |
|---|---|---|---|---|---|---|---|---|---|---|---|---|
| | Cells | Area | Power (mW) | WNS (ns) | TNS (ns) | Check | Cells | Area | Power (mW) | WNS (ns) | TNS (ns) | Check |
| adder | - | - | - | - | - | ✗ | - | - | - | - | - | ✗✗ |
| adder_select | 450 | 439.17 | 264.34 | 0.00 | 0.00 | ✔ | 450 | 439.17 | 264.34 | 0.00 | 0.00 | ✔ |
| alu_64bit | 1683 | 1645.21 | 887.78 | 0.00 | 0.00 | ✔ | 1683 | 1645.21 | 887.78 | 0.00 | 0.00 | ✔ |
| alu_8bit | 146 | 179.55 | 79.58 | 0.00 | 0.00 | ✔ | 146 | 179.55 | 79.58 | 0.00 | 0.00 | ✔ |
| calculation | - | - | - | - | - | ✗ | 888 | 1044.32 | 788.67 | 0.61 | 5.66 | ✔ |
| comparator | 59 | 52.67 | 26.44 | 0.00 | 0.00 | ✔ | 54 | 47.61 | 24.64 | 0.00 | 0.00 | ✔ |
| comparator_16bit | - | - | - | - | - | ✗ | 102 | 88.31 | 44.57 | 0.00 | 0.00 | ✔ |
| comparator_2bit | 10 | 9.58 | 4.51 | 0.00 | 0.00 | ✔ | 10 | 8.78 | 3.82 | 0.00 | 0.00 | ✔ |
| comparator_4bit | 21 | 18.35 | 9.25 | 0.00 | 0.00 | ✔ | 21 | 18.35 | 9.25 | 0.00 | 0.00 | ✔ |
| comparator_8bit | - | - | - | - | - | ✗ | 48 | 43.36 | 18.56 | 0.00 | 0.00 | ✔ |
| decoder_6bit | 86 | 71.29 | 19.76 | 0.00 | 0.00 | ✔ | 87 | 71.55 | 19.54 | 0.00 | 0.00 | ✔ |
| decoder_8bit | 312 | 246.85 | 50.28 | 0.00 | 0.00 | ✔ | 308 | 246.85 | 49.75 | 0.00 | 0.00 | ✔ |
| divider_16bit | 3445 | 3377.67 | 5030 | -3.32 | -73.2 | ✔ | 2461 | 2372.72 | 3750 | 7.22 | 156. | ✔ |
| divider_32bit | 14400 | 14295.90 | 23800 | 10.19 | 468. | ✔ | N/A | N/A | N/A | N/A | N/A | ✔ |
| divider_4bit | N/A | N/A | N/A | N/A | N/A | ✔ | - | - | - | - | - | ✗ |
| divider_8bit | 571 | 575.36 | 681.84 | 0.50 | 4.57 | ✔ | 266 | 264.14 | 188.61 | 0.51 | 3.16 | ✔ |
| fsm | 89 | 128.74 | 92.56 | 0.00 | 0.00 | ✔ | - | - | - | - | - | ✗✗ |
| fsm_encode | 287 | 416.82 | 337.09 | 0.00 | 0.00 | ✔ | 151 | 302.18 | 273.32 | 0.00 | 0.00 | ✔ |
| gray | - | - | - | - | - | ✗ | 51 | 69.69 | 66.78 | 0.00 | 0.00 | ✔ |
| mac | - | - | - | - | - | ✗ | - | - | - | - | - | ✗ |
| mul | 315 | 378.52 | 421.67 | 0.05 | 0.13 | ✔ | 315 | 378.52 | 421.67 | 0.05 | 0.13 | ✔ |
| mul_sub | 233 | 337.02 | 256.53 | 0.00 | 0.00 | ✔ | 234 | 341.81 | 255.66 | 0.00 | 0.00 | ✔ |
| mux | N/A | N/A | N/A | N/A | N/A | ✔ | N/A | N/A | N/A | N/A | N/A | ✔ |
| mux_encode | - | - | - | - | - | ✗ | N/A | N/A | N/A | N/A | N/A | ✗ |
| saturating_add | - | - | - | - | - | ✗ | 18 | 67.56 | 65.62 | 0.00 | 0.00 | ✔ |
| selector | 21 | 44.69 | 39.6 | 0.00 | 0.00 | ✔ | 18 | 39.37 | 36.06 | 0.00 | 0.00 | ✔ |
| sub_16bit | 132 | 136.19 | 77.92 | 0.00 | 0.00 | ✔ | 132 | 136.19 | 77.92 | 0.00 | 0.00 | ✔ |
| sub_32bit | 278 | 252.17 | 146.9 | 0.00 | 0.00 | ✔ | 278 | 252.17 | 146.9 | 0.00 | 0.00 | ✔ |
| sub_4bit | N/A | N/A | N/A | N/A | N/A | ✔ | - | - | - | - | - | ✗ |
| sub_8bit | 27 | 41.50 | 23.91 | 0.00 | 0.00 | ✔ | - | - | - | - | - | ✗ |
| add_sub | 124 | 130.34 | 101.74 | 0.00 | 0.00 | ✔ | 124 | 130.34 | 101.74 | 0.00 | 0.00 | ✔ |
| addr_calcu | 78 | 131.40 | 96.07 | 0.03 | 0.06 | ✔ | - | - | - | - | - | ✗ |
| mult_if | 10 | 10.91 | 3.53 | 0.00 | 0.00 | ✔ | 10 | 10.91 | 3.53 | 0.00 | 0.00 | ✔ |
| mux_large | 81 | 96.82 | 40.84 | 0.00 | 0.00 | ✔ | - | - | - | - | - | ✗ |
| register | 4625 | 9226.74 | 7540 | 0.00 | 0.00 | ✔ | - | - | - | - | - | ✗✗ |
| ticket_machine | - | - | - | - | - | ✗ | - | - | - | - | - | ✗ |

| Category | Deepseek V3 | | | | | | Deepseek R1 | | | | | |
|---|---|---|---|---|---|---|---|---|---|---|---|---|
| | Cells | Area | Power (mW) | WNS (ns) | TNS (ns) | Check | Cells | Area | Power (mW) | WNS (ns) | TNS (ns) | Check |
| adder | 433 | 511.52 | 368.04 | 0.00 | 0.00 | ✔ | 433 | 511.52 | 368.04 | 0.00 | 0.00 | ✔ |
| adder_select | 450 | 439.17 | 264.34 | 0.00 | 0.00 | ✔ | 306 | 350.06 | 225.74 | 0.00 | 0.00 | ✔ |
| alu_64bit | 1683 | 1645.21 | 887.78 | 0.00 | 0.00 | ✔ | 1697 | 1658.24 | 896.08 | 0.00 | 0.00 | ✔ |
| alu_8bit | 146 | 181.94 | 79.22 | 0.00 | 0.00 | ✔ | 146 | 179.55 | 79.58 | 0.00 | 0.00 | ✗ |
| calculation | 888 | 1044.32 | 788.67 | 0.61 | 5.66 | ✔ | 755 | 859.98 | 617.26 | 0.60 | 4.83 | ✔ |
| comparator | - | - | - | - | - | ✗ | 46 | 42.29 | 20.87 | 0.00 | 0.00 | ✔ |
| comparator_16bit | - | - | - | - | - | ✗ | 70 | 65.70 | 30 | 0.00 | 0.00 | ✔ |
| comparator_2bit | 23 | 21.28 | 10.86 | 0.00 | 0.00 | ✔ | 9 | 7.45 | 3.75 | 0.00 | 0.00 | ✔ |
| comparator_4bit | 46 | 44.42 | 22.8 | 0.00 | 0.00 | ✔ | - | - | - | - | - | ✗✗ |
| comparator_8bit | 48 | 42.29 | 20.42 | 0.00 | 0.00 | ✔ | - | - | - | - | - | ✗✗ |
| decoder_6bit | 87 | 71.55 | 19.54 | 0.00 | 0.00 | ✔ | 96 | 76.61 | 22.56 | 0.00 | 0.00 | ✗ |
| decoder_8bit | 312 | 246.85 | 50.28 | 0.00 | 0.00 | ✔ | 329 | 267.33 | 54.11 | 0.00 | 0.00 | ✔ |
| divider_16bit | - | - | - | - | - | ✗ | 3445 | 3377.67 | 5030 | 3.32 | 73.2 | ✔ |
| divider_32bit | - | - | - | - | - | ✗ | - | - | - | - | - | ✗✗ |
| divider_4bit | - | - | - | - | - | ✗ | 39 | 40.43 | 21.58 | 0.00 | 0.00 | ✗ |
| divider_8bit | - | - | - | - | - | ✗ | - | - | - | - | - | ✗✗ |
| fsm | 76 | 94.70 | 50.09 | 0.00 | 0.00 | ✔ | 73 | 92.04 | 51.44 | 0.00 | 0.00 | ✔ |
| fsm_encode | 138 | 294.20 | 305.96 | 0.00 | 0.00 | ✔ | 151 | 302.18 | 273.32 | 0.00 | 0.00 | ✔ |
| gray | 51 | 69.69 | 66.78 | 0.00 | 0.00 | ✔ | 52 | 70.22 | 65.86 | 0.00 | 0.00 | ✔ |
| mac | - | - | - | - | - | ✗ | 352 | 632.02 | 787.88 | 0.00 | 0.00 | ✗ |
| mul | - | - | - | - | - | ✗ | 315 | 378.52 | 421.67 | 0.05 | 0.13 | ✔ |
| mul_sub | 241 | 344.20 | 261.28 | 0.00 | 0.00 | ✔ | 234 | 336.22 | 259.8 | 0.00 | 0.00 | ✔ |
| mux | N/A | N/A | N/A | N/A | N/A | ✔ | N/A | N/A | N/A | N/A | N/A | ✔ |
| mux_encode | - | - | - | - | - | ✗ | 61 | 73.42 | 35.39 | 0.00 | 0.00 | ✔ |
| saturating_add | 18 | 67.56 | 65.77 | 0.00 | 0.00 | ✔ | - | - | - | - | - | ✗✗ |
| selector | 18 | 39.37 | 36.06 | 0.00 | 0.00 | ✔ | 18 | 39.37 | 36.06 | 0.00 | 0.00 | ✔ |
| sub_16bit | 122 | 126.88 | 71.97 | 0.00 | 0.00 | ✔ | 122 | 126.88 | 71.97 | 0.00 | 0.00 | ✔ |
| sub_32bit | 278 | 252.17 | 146.9 | 0.00 | 0.00 | ✔ | - | - | - | - | - | ✗ |
| sub_4bit | 14 | 19.42 | 10.61 | 0.00 | 0.00 | ✔ | 14 | 19.42 | 10.61 | 0.00 | 0.00 | ✔ |
| sub_8bit | 27 | 41.50 | 23.91 | 0.00 | 0.00 | ✔ | 27 | 41.50 | 23.91 | 0.00 | 0.00 | ✔ |
| add_sub | 124 | 130.34 | 101.74 | 0.00 | 0.00 | ✔ | - | - | - | - | - | ✗ |
| addr_calcu | 82 | 125.55 | 90.27 | 0.01 | 0.01 | ✔ | - | - | - | - | - | ✗ |
| mult_if | - | - | - | - | - | ✗ | - | - | - | - | - | ✗ |
| mux_large | 89 | 100.02 | 50.57 | 0.00 | 0.00 | ✔ | 89 | 100.02 | 50.57 | 0.00 | 0.00 | ✔ |
| register | - | - | - | - | - | ✗ | 3731 | 8745.55 | 7710 | 0.00 | 0.00 | ✔ |
| ticket_machine | - | - | - | - | - | ✗ | - | - | - | - | - | ✗ |

Table 6: PPA quality (DC *compile_ultra, 1ns*) and functional correctness for all designs optimized by GPT-4o-mini, Gemini-2.5, Deepseek V3, and Deepseek R1, using the RTL-OPT benchmark.

## D.2 PPA AND FUNCTIONAL CORRECTNESS OF LLM-OPTIMIZED DESIGNS (DC COMPILE, 1 NS)

Table 7 summarizes the PPA results and functional correctness checks for designs optimized by GPT-4o-mini, Gemini-2.5, Deepseek V3, and Deepseek R1 on the RTL-OPT benchmark. All evaluations are conducted under DC compile with 0.1ns clock period. This supplements Table 6, which uses a more relaxed timing setup.

| Category | GPT-4o-mini | | | | | | Gemini-2.5 | | | | | |
|---|---|---|---|---|---|---|---|---|---|---|---|---|
| | Cells | Area | Power (mW) | WNS (ns) | TNS (ns) | Check | Cells | Area | Power (mW) | WNS (ns) | TNS (ns) | Check |
| adder | - | - | - | - | - | ✗✗ | - | - | - | - | - | ✗✗ |
| adder_select | 813 | 812.1 | 4.85 | -0.39 | -10.8 | ✓ | 813 | 812.10 | 4.85 | -0.39 | -10.8 | ✓ |
| alu_64bit | 3248 | 3028 | 16.95 | -0.51 | -29.7 | ✓ | 3248 | 3028 | 16.95 | -0.51 | -29.7 | ✓ |
| alu_8bit | 402 | 370 | 1.89 | -0.26 | -1.91 | ✓ | 402 | 370.01 | 1.89 | -0.26 | -1.91 | ✓ |
| calculation | - | - | - | - | - | ✗ | 670 | 997.50 | 6.59 | -3.22 | -42.6 | ✓ |
| comparator | 96 | 83.26 | 0.38 | -0.22 | -0.22 | ✓ | 100 | 88.05 | 0.38 | -0.22 | -0.22 | ✓ |
| comparator_16bit | - | - | - | - | - | ✗ | 296 | 244.45 | 1.06 | -0.18 | -0.52 | ✓ |
| comparator_2bit | 14 | 14.36 | 0.06 | -0.02 | -0.05 | ✓ | 16 | 13.03 | 0.05 | -0.01 | -0.02 | ✓ |
| comparator_4bit | 33 | 29.26 | 0.18 | -0.05 | -0.15 | ✓ | 29 | 23.14 | 0.09 | -0.10 | -0.22 | ✓ |
| comparator_8bit | - | - | - | - | - | ✗ | 79 | 83.52 | 0.37 | -0.11 | -0.31 | ✓ |
| decoder_6bit | 128 | 109.9 | 0.31 | -0.03 | -1.74 | ✓ | 204 | 208.01 | 0.37 | -0.18 | -9.22 | ✓ |
| decoder_8bit | 781 | 855.9 | 1.28 | -0.27 | -58.8 | ✓ | 354 | 297.4 | 0.66 | -0.11 | -24.8 | ✓ |
| divider_16bit | 5037 | 5045 | 66.25 | -3.88 | -89.4 | ✓ | 3926 | 3895 | 51.75 | -6.85 | -155 | ✓ |
| divider_32bit | 16638 | 16211 | 171.11 | -13.23 | -846.3 | ✓ | 14529 | 13375 | 154.6 | -13.24 | -629 | ✓ |
| divider_4bit | 85 | 84.85 | 0.48 | -0.32 | -1.29 | ✓ | - | - | - | - | - | ✗ |
| divider_8bit | 744 | 731.8 | 6.94 | -1.21 | -13.5 | ✓ | 531 | 513.38 | 3.58 | -1.15 | -11.0 | ✓ |
| fsm | 149 | 192.8 | 1.06 | -0.28 | -4.40 | ✓ | - | - | - | - | - | ✗✗ |
| fsm_encode | 389 | 504.9 | 3.69 | -0.41 | -12.5 | ✓ | 357 | 448.21 | 3.16 | -0.40 | -7.95 | ✓ |
| gray | - | - | - | - | - | ✗ | 100 | 109 | 0.92 | -0.28 | -2.04 | ✓ |
| mac | - | - | - | - | - | ✗ | - | - | - | - | - | ✗ |
| mul | 311 | 453 | 3.64 | -1.26 | -9.69 | ✓ | 311 | 453 | 3.64 | -1.26 | -9.69 | ✓ |
| mul_sub | 634 | 623.2 | 4.17 | -0.75 | -8.08 | ✓ | 282 | 381.71 | 2.67 | -1.11 | -11.6 | ✓ |
| mux | 32 | 25.54 | 0.08 | 0.0 | 0.0 | ✓ | 32 | 25.54 | 0.08 | 0.0 | 0.0 | ✓ |
| mux_encode | - | - | - | - | - | ✗ | - | - | - | - | - | ✗ |
| saturating_add | - | - | - | - | - | ✗✗ | 127 | 140.98 | 1.08 | -0.36 | -2.86 | ✓ |
| selector | 30 | 53.73 | 0.48 | -0.18 | -0.49 | ✓ | 38 | 56.39 | 0.44 | -0.11 | -0.38 | ✓ |
| sub_16bit | 266 | 230.6 | 1.22 | -0.27 | -3.40 | ✓ | 266 | 230.62 | 1.22 | -0.27 | -3.40 | ✓ |
| sub_32bit | 551 | 476.4 | 2.57 | -0.35 | -9.13 | ✓ | 551 | 476.41 | 2.57 | -0.35 | -9.13 | ✓ |
| sub_4bit | 50 | 49.48 | 0.23 | -0.16 | -0.59 | ✓ | - | - | - | - | - | ✗ |
| sub_8bit | 121 | 102.7 | 0.51 | -0.23 | -1.30 | ✓ | - | - | - | - | - | ✗ |
| add_sub | 496 | 444.22 | 2.57 | -0.31 | -4.20 | ✓ | 496 | 444.22 | 2.57 | -0.31 | -4.20 | ✓ |
| addr_calcu | 405 | 388.09 | 3.07 | -0.62 | -8.56 | ✓ | - | - | - | - | - | ✗ |
| mult_if | 17 | 15.96 | 0.05 | -0.05 | -0.05 | ✓ | 17 | 15.96 | 0.05 | -0.05 | -0.05 | ✓ |
| mux_large | 296 | 273.45 | 0.84 | -0.11 | -0.85 | ✓ | - | - | - | - | - | ✗ |
| register | 4615 | 9480.24 | 78.21 | -0.36 | -251 | ✓ | - | - | - | - | - | ✗✗ |
| ticket_machine | - | - | - | - | - | ✗ | - | - | - | - | - | ✗ |

| Category | Deepseek V3 | | | | | | Deepseek R1 | | | | | |
|---|---|---|---|---|---|---|---|---|---|---|---|---|
| | Cells | Area | Power (mW) | WNS (ns) | TNS (ns) | Check | Cells | Area | Power (mW) | WNS (ns) | TNS (ns) | Check |
| adder | 530 | 618.7 | 4.51 | -0.43 | -12.5 | ✓ | 597 | 669.8 | 4.78 | -0.41 | -11.8 | ✓ |
| adder_select | 813 | 812.1 | 4.85 | -0.39 | -10.8 | ✓ | 514 | 522.7 | 3.49 | -0.42 | -12.0 | ✓ |
| alu_64bit | 3248 | 3028 | 16.95 | -0.51 | -29.7 | ✓ | 3248 | 3028 | 16.95 | -0.51 | -29.7 | ✓ |
| alu_8bit | 402 | 370 | 1.89 | -0.26 | -1.91 | ✓ | 402 | 370 | 1.89 | -0.26 | -1.91 | ✗ |
| calculation | 670 | 997.5 | 6.59 | -3.2 | -42.6 | ✓ | 550 | 808.1 | 5.73 | 3.29 | 44.8 | ✓ |
| comparator | - | - | - | - | - | ✗ | 110 | 96.29 | 0.4 | -0.19 | -0.19 | ✓ |
| comparator_16bit | - | - | - | - | - | ✗ | 122 | 107.2 | 0.48 | -0.17 | -0.48 | ✓ |
| comparator_2bit | 10 | 9.84 | 0.04 | -0.03 | -0.03 | ✓ | 13 | 11.17 | 0.04 | -0.03 | -0.04 | ✓ |
| comparator_4bit | 25 | 25.80 | 0.11 | -0.06 | -0.15 | ✓ | - | - | - | - | - | ✗✗ |
| comparator_8bit | 73 | 71.29 | 0.31 | -0.12 | -0.33 | ✓ | - | - | - | - | - | ✗✗ |
| decoder_6bit | 204 | 208.01 | 0.37 | -0.18 | -9.22 | ✓ | - | - | - | - | - | ✗ |
| decoder_8bit | 781 | 856 | 1.28 | -0.27 | -58.8 | ✓ | 392 | 330.1 | 0.72 | -0.10 | -23.1 | ✓ |
| divider_16bit | - | - | - | - | - | ✗ | 5037 | 5045 | 66.25 | -3.88 | -89.4 | ✓ |
| divider_32bit | - | - | - | - | - | ✗ | - | - | - | - | - | ✗✗ |
| divider_4bit | - | - | - | - | - | ✗ | - | - | - | - | - | ✗ |
| divider_8bit | - | - | - | - | - | ✗ | - | - | - | - | - | ✗✗ |
| fsm | 100 | 129.5 | 0.6 | -0.22 | -2.62 | ✓ | 106 | 129.8 | 0.59 | -0.24 | -2.70 | ✓ |
| fsm_encode | 372 | 466.8 | 3.46 | -0.38 | -9.36 | ✓ | 357 | 448.2 | 3.16 | -0.40 | -7.95 | ✓ |
| gray | 100 | 109.1 | 0.92 | -0.28 | -2.04 | ✓ | 100 | 109.1 | 0.92 | -0.28 | -2.04 | ✓ |
| mac | - | - | - | - | - | ✗ | - | - | - | - | - | ✗ |
| mul | - | - | - | - | - | ✗ | 270 | 426.4 | 3.43 | -1.25 | -9.62 | ✓ |
| mul_sub | 657 | 654.4 | 4.44 | -0.76 | -8.08 | ✓ | 675 | 654.1 | 4.46 | -0.70 | -7.97 | ✓ |
| mux | 32 | 25.54 | 0.08 | 0.0 | 0.0 | ✓ | 32 | 25.54 | 0.82 | 0.0 | 0.0 | ✓ |
| mux_encode | - | - | - | - | - | ✗ | 113 | 121.6 | 0.46 | -0.09 | -0.69 | ✓ |
| saturating_add | 127 | 140.9 | 1.08 | -0.36 | -2.86 | ✓ | - | - | - | - | - | ✗ |
| selector | 38 | 56.39 | 0.44 | -0.11 | -0.38 | ✓ | 30 | 49.74 | 0.41 | -0.18 | -0.49 | ✓ |
| sub_16bit | 266 | 230.6 | 1.22 | -0.27 | -3.40 | ✓ | 266 | 230.6 | 1.22 | -0.27 | -3.40 | ✓ |
| sub_32bit | 551 | 476.4 | 2.57 | -0.35 | -9.13 | ✓ | - | - | - | - | - | ✗ |
| sub_4bit | 37 | 37.24 | 0.18 | -0.10 | -0.30 | ✓ | 37 | 37.24 | 0.12 | -0.10 | -0.30 | ✓ |
| sub_8bit | 121 | 102.7 | 0.51 | -0.23 | -1.30 | ✓ | 121 | 102.7 | 0.51 | -0.23 | -1.30 | ✓ |
| add_sub | 496 | 444.2 | 2.57 | -0.31 | -4.20 | ✓ | - | - | - | - | - | ✗ |
| addr_calcu | 229 | 214.40 | 1.8 | -0.60 | -8.47 | ✓ | - | - | - | - | - | ✗ |
| mult_if | - | - | - | - | - | ✗ | - | - | - | - | - | ✗ |
| mux_large | 173 | 174.5 | 0.62 | -0.12 | -0.95 | ✓ | 173 | 174.5 | 0.62 | -0.12 | -0.95 | ✓ |
| register | - | - | - | - | - | ✗ | 5003 | 9780 | 79.63 | -0.40 | -245 | ✓ |
| ticket_machine | - | - | - | - | - | ✗ | - | - | - | - | - | ✗ |

Table 7: PPA quality (DC *compile, 0.1ns*) and functional correctness for all designs optimized by GPT-4o-mini, Gemini-2.5, Deepseek V3, and Deepseek R1, using the RTL-OPT benchmark.

# E  DETAILED LLM EVALUATION RESULTS AND STATISTICAL ANALYSIS

We **re-run each LLM evaluation 5 times** on RTL-OPT (36 designs), using the same prompt template to ensure consistency. For each run, we recorded the: (1) Syntax correctness; (2) Func correctness; (3) PPA better than suboptimal; (4) PPA better than optimized. The experimental results for 6 different LLMs (**including two open-source LLMs**) are summarized below.

We summarize statistical results and conduct paired t-tests on two representative LLM pairs: Gmini vs. DS-R1 and LLaMA vs. Qwen. We will include the complete **mean ± $\sigma$** and **paired t-tests** results in this appendix when updating the camera-ready version.

## E.1  SYNTAX CORRECTNESS (N OUT OF 36)

| Run | GPT-4o | Gmini-2.5 | DS V3 | DS R1 | QWEN | Llama |
|---|---|---|---|---|---|---|
| 1 | 36 | 35 | 36 | 32 | 34 | 15 |
| 2 | 35 | 31 | 34 | 32 | 34 | 17 |
| 3 | 36 | 30 | 36 | 31 | 31 | 15 |
| 4 | 35 | 34 | 36 | 30 | 30 | 14 |
| 5 | 35 | 35 | 36 | 32 | 34 | 15 |
| **Mean** | 35.4 | 33 | 35.6 | 31.4 | 32.6 | 15.2 |
| $\sigma$ | 1.0 | 1.5 | 1.3 | 0.8 | 0.9 | 1.1 |

## E.2  FUNCTIONAL CORRECTNESS (N OUT OF 36)

| Run | GPT-4o | Gmini-2.5 | DS V3 | DS R1 | QWEN | Llama |
|---|---|---|---|---|---|---|
| 1 | 29 | 26 | 27 | 24 | 23 | 13 |
| 2 | 28 | 25 | 21 | 26 | 20 | 18 |
| 3 | 25 | 24 | 26 | 25 | 23 | 17 |
| 4 | 27 | 25 | 25 | 24 | 21 | 12 |
| 5 | 28 | 28 | 25 | 26 | 19 | 17 |
| **Mean** | 27.4 | 25.6 | 24.8 | 25 | 21.2 | 15.4 |
| $\sigma$ | 1.5 | 1.5 | 2.3 | 1.8 | 1.6 | 2.7 |

## E.3  PPA >SUBOPTIMAL (N OUT OF 36)

| Run | GPT-4o | Gmini-2.5 | DS V3 | DS R1 | QWEN | Llama |
|---|---|---|---|---|---|---|
| 1 | 7 | 11 | 10 | 15 | 4 | 2 |
| 2 | 8 | 9 | 6 | 16 | 4 | 5 |
| 3 | 7 | 10 | 8 | 16 | 6 | 4 |
| 4 | 7 | 9 | 10 | 13 | 4 | 1 |
| 5 | 9 | 10 | 8 | 14 | 4 | 2 |
| **Mean** | 7.6 | 9.8 | 8.4 | 14.8 | 4.4 | 2.8 |
| $\sigma$ | 0.9 | 0.8 | 1.5 | 1.3 | 1.1 | 1.6 |

## E.4  PPA >OPTIMIZED (N OUT OF 36)

| Run | GPT-4o | Gmini-2.5 | DS V3 | DS R1 | QWEN | Llama |
|---|---|---|---|---|---|---|
| 1 | 2 | 2 | 5 | 5 | 1 | 0 |
| 2 | 3 | 2 | 4 | 6 | 0 | 1 |
| 3 | 1 | 4 | 5 | 5 | 2 | 1 |
| 4 | 3 | 2 | 5 | 5 | 2 | 0 |
| 5 | 2 | 2 | 3 | 4 | 1 | 1 |
| **Mean** | 2.2 | 2.4 | 4.4 | 5 | 1.2 | 0.6 |
| $\sigma$ | 0.8 | 0.9 | 1.0 | 0.7 | 0.8 | 0.5 |

## E.5 Comparison: GPT-4o vs. DS-R1

| Metric | Mean Diff | t-value | p-value | Significance | Effect Size |
|---|---|---|---|---|---|
| Syntax correctness | +3.8 | +6.53 | 0.0028 | Significant | 2.92 (Very Large) |
| Func correctness | +1.6 | +1.67 | 0.169 | Not Significant | 0.75 (Medium) |
| PPA >suboptimal | -7.2 | -5.81 | 0.004 | Significant | 2.60 (Very Large) |
| PPA >optimized | -2.8 | -5.83 | 0.004 | Significant | 2.61 (Very Large) |

## E.6 Comparison: LLaMA vs. Qwen

| Metric | Mean Diff | t-value | p-value | Significance | Effect Size |
|---|---|---|---|---|---|
| Syntax correctness | -18.2 | -48.40 | <0.0001 | Significant | -21.7 (Extreme) |
| Func correctness | -5.8 | -5.39 | 0.006 | Significant | -2.41 (Very Large) |
| PPA >suboptimal | +1.6 | +2.53 | 0.065 | Marginal | 0.72 (Medium) |
| PPA >optimized | +0.6 | +1.34 | 0.251 | Not Significant | 0.30 (Small) |

# F  Qualitative Analysis of LLM Functionality Failures

As summarized in Section 4, LLMs face a fundamental trade-off: conservative approaches avoid errors but miss optimization opportunities, while aggressive optimizations risk introducing functional bugs despite correct syntax.

To systematically analyze failure modes, we conducted a detailed study of 40 randomly sampled cases where LLM-generated optimizations passed syntax checks but introduced functional errors. **Our findings reveal three LLM failure categories**:

- **Control Logic Inconsistencies** (19/40, 47.5%)

    Example (`comparator`): LLM failed to maintain bit-priority ordering (MSB-first comparison) and incorrectly implemented the 'lt' condition using wrong Boolean logic.

- **Overly Aggressive Pipelining** (12/40, 30%)

    Example (`fsm`): LLM reduced pipeline stages from 4 to 3 cycles, violating the original design's latency requirements and causing incorrect output timing.

- **Improper Resource Sharing** (9/40, 22.5%)

    Example (`mux`): Register sharing ignored temporal dependencies, leading to stale data being read in subsequent cycles.

# G  Reproduction of Existing Benchmark with Multiple Synthesis Flows

## G.1  Reproduction of (Yao et al., 2024) Designs with Multiple Synthesis Flows

Table 8 provides reproduction results for 14 designs originally optimized by (Yao et al., 2024). Both the baseline (original) and expert-optimized versions are evaluated using our unified flow, across three synthesis settings: Yosys, DC compile (0.1ns), and DC compile ultra (1ns). This ensures fair comparisons across benchmarks and demonstrates the effectiveness of our pipeline in capturing prior work.

| Design List | Yosys | | DC (compile, 0.1ns) | | | | | DC (compile_ultra, 1ns) | | | | |
|---|---|---|---|---|---|---|---|---|---|---|---|---|
| | Cells | Area | Cells | Area | Power | WNS | TNS | Cells | Area | Power | WNS | TNS |
| case1 | 18 | 57.99 | - | - | - | - | - | - | - | - | - | - |
| case1_opt | 12 | 30.86 | - | - | - | - | - | - | - | - | - | - |
| case2 | 37 | 44.95 | - | - | - | - | - | - | - | - | - | - |
| case2_opt | 37 | 44.95 | - | - | - | - | - | - | - | - | - | - |
| case3 | 107 | 129.3 | 289 | 255.1 | 2.25 mW | -0.41 | -3.23 | 26 | 98.95 | 74.85 uW | 0.0 | 0.0 |
| case3_opt | 102 | 127.7 | 289 | 255.1 | 2.25 mW | -0.41 | -3.23 | 26 | 98.95 | 74.85 uW | 0.0 | 0.0 |
| case4 | 304 | 347.1 | 576 | 550.9 | 3.75 mW | -0.51 | -3.72 | 204 | 267.06 | 169.5 uW | 0.0 | 0.0 |
| case4_opt | 262 | 310.7 | 553 | 529.9 | 3.80 mW | -0.50 | -3.65 | 233 | 291.27 | 165.6 uW | 0.0 | 0.0 |
| case5 | 10933 | 12457 | 6546 | 11104 | 173.7 mW | -4.15 | -413.6 | 6047 | 9668 | 19.99 mW | -1 | -56.61 |
| case5_opt | 10504 | 11980 | 6044 | 10469 | 168.9 mW | -4.22 | -418.1 | 5962 | 9600 | 19.98 mW | -1.02 | -56.7 |
| case6 | 37 | 59.32 | 53 | 60.38 | 484.8 uW | -0.22 | -0.68 | 28 | 41.76 | 42.93 uW | 0.0 | 0.0 |
| case6_opt | 21 | 28.99 | 23 | 28.99 | 284.1 uW | -0.15 | -0.28 | 11 | 19.95 | 23.42 uW | 0.0 | 0.0 |
| case7 | 24 | 38.84 | 22 | 34.31 | 418.7 uW | -0.15 | -0.46 | 16 | 28.99 | 40.65 uW | 0.0 | 0.0 |
| case7_opt | 11 | 19.95 | 11 | 18.35 | 215.8 uW | -0.12 | -0.25 | 8 | 15.96 | 21.90 uW | 0.0 | 0.0 |
| case8 | 1471 | 1720 | 1353 | 2242 | 13.90 mW | -0.92 | -58.2 | 703 | 831.2 | 634.7 uW | 0.0 | 0.0 |
| case8_opt | 661 | 758.1 | 1771 | 1962 | 14.87 mW | -0.95 | -57.0 | 711 | 834.7 | 647.3 uW | 0.0 | 0.0 |
| case9 | 1716 | 2010 | 1336 | 2420 | 13.31 mW | -1.05 | -65.7 | 729 | 886.3 | 636.4 uW | 0.0 | 0.0 |
| case9_opt | 776 | 926.5 | 1757 | 1984 | 16.50 mW | -1.00 | -70.6 | 973 | 965.1 | 982.7 uW | 0.0 | 0.0 |
| case10 | 24 | 43.89 | 25 | 31.92 | 103.8 uW | -0.02 | -0.14 | 25 | 31.92 | 10.38 uW | 0.0 | 0.0 |
| case10_opt | 24 | 8.51 | 40 | 38.30 | 114.8 uW | -0.01 | -0.07 | 25 | 21.81 | 8.38 uW | 0.0 | 0.0 |
| case11 | 2 | 1.86 | 9 | 5.32 | 16.07 uW | 0.0 | 0.0 | 1 | 1.06 | 252.5 nW | 0.0 | 0.0 |
| case11_opt | 1 | 1.86 | 6 | 2.93 | 7.28 uW | 0.0 | 0.0 | 1 | 1.06 | 252.5 nW | 0.0 | 0.0 |
| case12 | 1 | 1.86 | 3 | 0.0 | 374.8 nW | 0.0 | 0.0 | 3 | 0.0 | 37.48 nW | 0.0 | 0.0 |
| case12_opt | 1 | 1.06 | 1 | 1.06 | 2.59 uW | 0.0 | 0.0 | 1 | 1.06 | 258.8 nW | 0.0 | 0.0 |
| case13 | 3 | 1.86 | 3 | 5.59 | 16.93 uW | -0.03 | -0.03 | 5 | 5.05 | 1.77 uW | 0.0 | 0.0 |
| case13_opt | 2 | 1.86 | 2 | 3.72 | 12.41 uW | -0.03 | -0.03 | 5 | 5.05 | 1.77 uW | 0.0 | 0.0 |
| case14 | 4 | 1.86 | 6 | 7.98 | 28.15 uW | -0.10 | -0.10 | 5 | 5.05 | 1.77 uW | 0.0 | 0.0 |
| case14_opt | 3 | 1.86 | 6 | 6.65 | 23.20 uW | -0.08 | -0.08 | 5 | 5.05 | 1.77 uW | 0.0 | 0.0 |

Table 8: Reproduction PPA results of 14 designs in (Yao et al., 2024) (both original and expert-optimized versions) using Yosys and DC synthesis flows.

## G.2 REPRODUCTION OF (YAO ET AL., 2024) GPT-OPTIMIZED DESIGNS

Table 9 and Table 10 provide the PPA reproduction of designs optimized by GPT-based methods and (Yao et al., 2024) own optimization strategies, respectively. These results help evaluate the generalizability of our flow when applied to designs outside the RTL-OPT benchmark and provide fair comparisons across different optimization methodologies. The same three synthesis settings are used (Yosys, DC compile 0.1ns, and DC compile ultra 1ns), under our unified evaluation flow.

| Design List | Yosys | | DC (compile, 0.1ns) | | | | | DC (compile_ultra, 1ns) | | | | |
|---|---|---|---|---|---|---|---|---|---|---|---|---|
| | Cells | Area | Cells | Area | Power | WNS | TNS | Cells | Area | Power | WNS | TNS |
| case1_GPT | 18 | 57.99 | 18 | 57.99 | 584.6 uW | -0.24 | -0.24 | 18 | 57.99 | 61.13 uW | 0.00 | 0.00 |
| case2_GPT | 37 | 44.95 | - | - | - | - | - | - | - | - | - | - |
| case3_GPT | 107 | 129.3 | 289 | 255.1 | 2.25 mW | -0.41 | -3.23 | 26 | 98.95 | 74.85 uW | 0.00 | 0.00 |
| case4_GPT | 290 | 326.6 | 637 | 602.8 | 4.04 mW | -0.47 | -3.33 | 205 | 265.5 | 159.5 uW | 0.00 | 0.00 |
| case5_GPT | 10627 | 12090 | 5987 | 10483 | 165.9 mW | -4.18 | -408 | 5962 | 9600 | 19.98 mW | -1.02 | -56.72 |
| case6_GPT | 37 | 59.32 | 50 | 59.85 | 492.7 uW | -0.25 | -0.73 | 25 | 38.57 | 40.19 uW | 0.00 | 0.00 |
| case7_GPT | 26 | 44.69 | 30 | 43.62 | 522.1 uW | -0.15 | -0.55 | 17 | 34.31 | 47.46 uW | 0.00 | 0.00 |
| case8_GPT | 844 | 952.3 | 2129 | 1915 | 15.04 mW | -0.67 | -44.17 | 704 | 932.1 | 722.7 uW | 0.00 | 0.00 |
| case9_GPT | 844 | 952.3 | 2129 | 1915 | 15.04 mW | -0.67 | -44.17 | 704 | 932.1 | 722.7 uW | 0.00 | 0.00 |
| case10_GPT | 24 | 21.38 | 32 | 25.54 | 81.96 uW | 0.0 | 0.0 | 25 | 21.81 | 8.382 uW | 0.00 | 0.00 |
| case11_GPT | 3 | 1.862 | 3 | 5.59 | 16.86 uW | -0.03 | -0.03 | 4 | 3.724 | 1.339 uW | 0.00 | 0.00 |
| case12_GPT | 1 | 1.064 | 1 | 1.06 | 2.59 uW | 0.0 | 0.0 | 1 | 1.064 | 258.8 nW | 0.00 | 0.00 |
| case13_GPT | 3 | 5.586 | 6 | 5.59 | 20.58 uW | 0.0 | 0.0 | 5 | 5.054 | 1.767 uW | 0.00 | 0.00 |
| case14_GPT | 2 | 3.724 | 5 | 4.26 | 16.58 uW | 0.0 | 0.0 | 4 | 3.724 | 1.248 uW | 0.00 | 0.00 |

Table 9: Reproduction PPA results of GPT-4 optimized designs in (Yao et al., 2024) using Yosys and DC synthesis flows.

| Design List | Yosys | | DC (compile, 0.1ns) | | | | | DC (compile_ultra, 1ns) | | | | |
|---|---|---|---|---|---|---|---|---|---|---|---|---|
| | Cells | Area | Cells | Area | Power | WNS | TNS | Cells | Area | Power | WNS | TNS |
| case1_ours | 14 | 39.9 | 13 | 40.96 | 431.3 uW | -0.21 | -0.28 | 14 | 39.90 | 44.67 uW | 0.0 | 0.0 |
| case2_ours | 37 | 44.95 | 76 | 69.43 | 383.9 uW | -0.17 | -1.07 | 9 | 32.45 | 15.31 uW | 0.0 | 0.0 |
| case3_ours | 107 | 129.3 | 289 | 255.1 | 2.251 mW | -0.41 | -3.23 | 26 | 98.95 | 74.85 uW | 0.0 | 0.0 |
| case4_ours | 168 | 178.8 | 396 | 371.9 | 2.212 mW | -0.44 | -3.46 | 153 | 163.32 | 110.87 uW | 0.0 | 0.0 |
| case5_ours | 10540 | 12016 | 6329 | 10642 | 170.7 mW | -4.14 | -409.54 | 5962 | 9600.21 | 19.98 mW | -1.02 | -56.72 |
| case6_ours | 20 | 28.73 | 23 | 28.99 | 284.1 uW | -0.15 | -0.28 | 11 | 19.95 | 23.42 uW | 0.0 | 0.0 |
| case7_ours | 23 | 35.64 | 23 | 34.58 | 385.6 uW | -0.14 | -0.47 | 14 | 27.40 | 35.59 uW | 0.0 | 0.0 |
| case8_ours | 908 | 1063 | 2136 | 1916 | 14.75 mW | -0.69 | -44.12 | 709 | 939.78 | 746.77 uW | 0.0 | 0.0 |
| case9_ours | 956 | 1173 | 2595 | 2354 | 19.54 mW | -0.95 | -61.83 | 847 | 1341.97 | 1.075 mW | 0.0 | 0.0 |
| case10_ours | - | - | - | - | - | - | - | - | - | - | - | |
| case11_ours | 1 | 1.862 | 7 | 4.788 | 15.60 uW | 0.0 | 0.0 | 2 | 1.330 | 370.2 nW | 0.0 | 0.0 |
| case12_ours | 1 | 1.062 | 1 | 1.064 | 2.588 uW | 0.0 | 0.0 | 1 | 1.064 | 258.8 nW | 0.0 | 0.0 |
| case13_ours | - | - | - | - | - | - | - | - | - | - | - | |
| case14_ours | 4 | 1.862 | 6 | 7.98 | 28.15 uW | -0.1 | -0.1 | 5 | 5.054 | 1.767 uW | 0.0 | 0.0 |

Table 10: Reproduction PPA results of RTLRewriter optimized their own designs using Yosys and DC synthesis flows.

# H  LLM Prompt Generation Process

The complete implementation of the LLM prompt generation and evaluation pipeline for the RTL-OPT benchmark is available in the anonymous repository. The core prompt generation process is implemented in Results/LLM_Test_result/llm_gen.py, (available in the anonymous repository).

