# OpenReview forum: "RTL-OPT: Rethinking the Generation of PPA-Optimized RTL Code and A New Benchmark"
_ICLR.cc/2026/Conference — ICLR 2026 Conference Withdrawn Submission_

### Official Review · Reviewer_E9bC · 2025-10-30

**Soundness:** 2
**Presentation:** 2
**Contribution:** 2
**Rating:** 2
**Confidence:** 3

**Summary:**

This paper introduces RTL-OPT, a new benchmark for evaluating the ability of large language models (LLMs) to optimize Register-Transfer Level (RTL) code for digital integrated circuit (IC) design, specifically focusing on improvements in power, performance, and area (PPA). The authors argue that existing benchmarks either focus solely on accuracy or contain contrived optimization scenarios that do not reflect genuine industrial challenges. RTL-OPT provides 36 expert-curated design tasks, each with paired suboptimal and optimized RTL codes, spanning diverse design types and real-world optimization patterns. An auto-evaluation framework is provided to verify correctness and quantify PPA improvements.

**Strengths:**

- RTL-OPT fills a gap by providing a carefully curated benchmark for RTL code optimization, targeting industrially relevant PPA metrics that current benchmarks neglect.
- The paper delivers a critical and thorough analysis of existing benchmarks and convincingly demonstrates their limitations through empirical comparison.
- The benchmark and toolchain are available for public use, supporting reproducibility and community adoption.

**Weaknesses:**

- With only 36 design tasks, the dataset is much smaller than many software-focused code benchmarks, limiting statistical robustness. This is acknowledged (Section 5), but the scalability challenge (manual curation) is unresolved.
- The designs are exclusively hand-designed by a limited pool of experts and reflect a subjective choice of optimization patterns. There is no principled selection or sampling strategy backing the coverage claims (Section 3.1), nor an attempt to quantify how representative or ‘real-world’ these patterns are with respect to the broader landscape of existing RTL code in the wild.
- The experimental comparison in Tables 6–7 and Figure 4 focuses on a handful of LLMs, all of which are relatively recent proprietary or publicly available models. There is almost no inclusion of classical, rule-based, or non-LLM optimization approaches as baselines, nor efforts to integrate domain-specific program transformation tools.
- The reporting and discussion around PPA improvements (Tables 4–5, Table 6–7) focus on absolute gain/loss counts and percentages. While PPA is inherently multidimensional (area, performance, power), the paper lacks more nuanced multi-objective analysis, such as Pareto fronts or weighted trade-off metrics.

**Questions:**

- Do the authors have a plan or methodology for scaling the benchmark beyond the current 36 tasks? Can bootstrapping with synthetic or crowdsourced designs address sample bias and improve statistical power?
- How did the authors select optimization patterns and design categories—was there any survey, mining of existing open-source designs, or engagement with industry partners to ensure breadth?
- Are the conclusions regarding LLM ranking statistically robust? Were statistical tests performed across multiple seeds/runs, or are results potentially sensitive to prompt engineering, seed, or synthesis setup?
- How is equivalence checking handled when timing/pipelining modifications change cycle-accurate behavior but maintain Black-box functionality? Are there borderline cases where formal or dynamic verification could disagree?

---

### Official Review · Reviewer_fNnC · 2025-10-30

**Soundness:** 3
**Presentation:** 3
**Contribution:** 1
**Rating:** 2
**Confidence:** 5

**Summary:**

The paper proposes RTL-OPT, a benchmark to evaluate whether LLMs can improve RTL code for power, performance, area (PPA). RTL-OPT comprises 36 hand-crafted RTL pairs spanning arithmetic, control/FSM, pipelined datapaths, etc.

**Strengths:**

This work dddress a real evaluation gap of LLMs in EDA: moves beyound code correctness to PPA optimization. Moreover, the author(s) provide a benchmark with 36 designs with the tool scripts and an automated evaluation pipeline.

**Weaknesses:**

1. Benchmark scale and representativeness are not enough. Many RTL-OPT circuits are small, in Table 4, most of circuits have fewer than ~100 cells. The larger designs (e.g., dividers) appear bigger due to for-loop unrolling. In terms of RTL lines of code and structural complexity, the tasks remain easy for modern LLMs.
2. Limited novelty compared with RTLRewriter. The main contribution is a benchmark and evaluation for LLM-based RTL code optimization. However, The 36 tasks are relatively simple and not clearly representative of real-world designs. In addition, the paper does not propose an efficient optimization methodology beyond evaluating base models on these tasks.

**Questions:**

1. I go through the evaluation framework. May I confirm if the synthesis command “compile_ultra” available? In my view, for small designs, no matter how the RTL code is written, the synthesized netlists produced by “compile_ultra” are essentially the same.
2. What’s the performance of domain-specific LLM, like ChipExpert[1].
[1] ChipExpert: The Open-Source Integrated-Circuit-Design-Specific Large Language Model

---

### Official Review · Reviewer_v2u6 · 2025-10-31

**Soundness:** 2
**Presentation:** 3
**Contribution:** 2
**Rating:** 2
**Confidence:** 4

**Summary:**

The paper introduces RTL-OPT, a benchmark with 36 handcrafted pairs of suboptimal and optimized RTL designs (Verilog) spanning arithmetic units, FSMs, control logic, and pipelined datapaths. It critiques prior RTL optimization benchmarks for contrived designs and synthesis sensitivity, provides an evaluation framework for functional equivalence and PPA (using Yosys/DC), and evaluates LLMs on the benchmark.

**Strengths:**

- Identifies a gap in evaluating LLMs for PPA-optimized RTL beyond mere functional correctness.
- The writing quality and illustrations are good, and the paper structure is also good.

**Weaknesses:**

- **Tiny scale**: Most designs are trivial (<100 cells post-synthesis per Table 4 in Appendix C); largest ~20K cells but only a few. Fails to represent real IC blocks (e.g., no multi-kilocycle datapaths or IP interfaces). Handcrafting 36 small toys isn't scalable or convincing for "industry-standard" claims.
- **Limited novelty**: Patterns (bit-width opt, resource sharing, etc.) are basic RTL tips from any FPGA/ASIC textbook (e.g., "Digital Design" by Harris). No novel optimizations; just repackaged common sense.
- **Overclaims**: "Handcrafted by human designers" + "proven industry practices" but designs too simplistic—real chips have 100s of interacting constraints . LLM evals in Sec 4 buried in appendix; weak results (e.g., GPT-4o wins 10/36) downplayed.
- **Eval flaws**: PPA trade-offs ignored in "better" counts; no power/dynamic timing paths; single library (assume Nangate45?); no multi-corner. Yosys-DC comparison good but cherry-picks clocks (0.1/1ns).

**Questions:**

- Why no medium-scale designs ? Handcrafting excuse weak—use open RTL + mutate realistically.
- Table 4: Confirm cell counts/library? Most ~50 cells = toy counters/adders, not benchmark-worthy.
- LLM results: Full pass@K or just zero-shot? Agents/prompts in appendix? Why no CodeLlama/DeepSeek-Coder?

---

### Official Review · Reviewer_X4bs · 2025-10-31

**Soundness:** 2
**Presentation:** 3
**Contribution:** 2
**Rating:** 4
**Confidence:** 4

**Summary:**

This paper argues that while the generation of PPA-optimized RTL code is important, most existing benchmarks focus primarily on the correctness of RTL generation. Only one existing benchmark (Yao et al., 2024) considers PPA-related issues. The authors' analysis suggests that this prior benchmark falls short in several aspects, including unrealistic designs, an oversimplified synthesis setup, and insufficient evaluation. To address these limitations, this paper introduces RTL-OPT, a benchmark for hardware RTL code optimization aimed at enhancing IC design quality. RTL-OPT includes 36 handcrafted digital IC designs, each provided with both suboptimal and optimized RTL code. An integrated evaluation framework is incorporated to verify functional correctness and quantify PPA improvements. The paper also provides a systematic analysis of RTL code optimization, covering the impact of synthesis, evaluations of existing and new benchmarks, and case studies. Finally, the authors present experiments evaluating the optimization capabilities of various LLMs on the RTL-OPT benchmark.

**Strengths:**

This paper proposes a new problem: despite growing interest in using LLMs for optimized RTL generation, only one inadequate benchmark currently exists. The authors' new benchmark tries to fill this gap. The work includes examination of related work and provides comprehensive comparisons with the proposed benchmark.

**Weaknesses:**

1. The authors note in the introduction that recent LLM research has begun targeting the generation of more optimized RTL code to improve PPA outcomes. However, the experimental section does not evaluate such works on the proposed RTL-OPT benchmark.
2. While the paper evaluates the commercial LLMs, such as GPT4, Gemini and DeepSeek, the experiment is not sufficient. The other general LLMs (e.g. Claude and Qwen) and customized LLMs (e.g. the LLMs for RTL generation) should be evaluated.
3. The paper highlights that the overall performance of all LLMs is not very good. However, this work does not provide any algorithm, framework or approach to address this issue.

**Questions:**

1. These circuits in RTL-OPT are very small, how do the LLMs perform on large-scale circuits?
2. For my understanding, current LLMs are still facing challenge to even generate highly accurate RTL code. Is it a bit too early for us to start paying attention to the performance of the RTL code generated by the model in terms of PPA now?

---

### Note · Authors · 2025-11-12

I have read and agree with the venue's withdrawal policy on behalf of myself and my co-authors.